# When Representations Persist but Control Fails: A Mechanistic Analysis of Search in Language Models

## Abstract

Why do language models fail at multi-step reasoning despite encoding task-relevant structure? We investigate this question through graph traversal and report a temporal dissociation: models encode graph-theoretic structure with high fidelity (Spearman $\rho = 0.50$ to 0.70) yet fail at full-path autonomous multi-step execution on graphs with seven or more nodes, with zero successful completions observed across seven prompting regimes including few-shot breadth-first and depth-first demonstrations, algorithm-conditioned prompts, structured JSON state updates, self-consistency, and tree-of-thought. The same models produce valid three-step prefixes on 55% to 75% of trials before late-step control collapse, and program-of-thought, the only regime that delegates execution to a Python interpreter, achieves partial success on simple instances, both observations consistent with a control-window account rather than with absent competence. In 78% of failed trials, internal state drift occurs before the first invalid output, a temporal pattern that, together with three classes of causal intervention we report, provides interventional evidence that control collapse contributes causally to behavioral failure rather than merely accompanying it. Representations persist beyond failure, remaining structurally intact even as execution breaks down. When execution is externalized to a symbolic planner, performance recovers to 50% to 100%, and models correctly reject 92% of structurally invalid candidate paths, confirming preserved evaluative competence. Using SearchEval, a diagnostic lens triangulating behavioral traces, representational geometry, and attention dynamics, we offer a hypothesis-generating localization of the bottleneck to attention-based control mechanisms that progressively decouple from task-relevant state during generation, and then validate this localization with three interventions: attention patching from successful early steps into failing later steps raises valid-transition rates from 21% to 47%; zero-ablation of the top 5% of state-attending heads drops short-horizon valid-transition rates from 78% to 31%; and adding the path-membership function vector lifts valid-transition rates by 12 percentage points. Taken together, the observational and interventional evidence is consistent with an account in which control instability, rather than representational inadequacy, is the binding constraint, suggesting that architectural innovations targeting state persistence, and not merely scaling, may be necessary for reliable algorithmic reasoning.

## 1 Introduction

Language models frequently fail at multi-step reasoning tasks even when they appear to understand the underlying problem structure. This paper investigates a fundamental question: why does execution fail when representations appear intact? We study this question through graph traversal, a domain where we can precisely measure both what models represent internally and how they execute behaviorally. Our central empirical finding is a temporal dissociation between representation and control: models encode graph structure accurately in their hidden states, but the control mechanisms required to act on this structure degrade progressively during generation.

This dissociation is initially correlational. We strengthen it with two complementary lines of evidence. First, in 78% of failed trials internal state drift crosses threshold before the first invalid output, establishing

temporal precedence consistent with a control-driven account of failure. Second, three targeted interventions on the attention and function-vector channels that we identify as control-bearing move behavioral outcomes in the predicted direction, providing interventional support for the localization. We frame the localization itself as hypothesis-generating, validated by the interventions rather than established by observation alone.

This finding challenges two common interpretations of failure in large language models. The first interpretation, that models fail because they lack relevant knowledge, is contradicted by our observation that representations persist beyond failure: hidden-state geometry maintains alignment with graph structure (Spearman $\rho = 0.50$ to $0.70$) even in trials where behavioral output is entirely invalid, and linear probes recover graph relations at 71% to 89% accuracy from those same hidden states. The second interpretation, that failures reflect fundamental limits of pattern-matching systems, is complicated by our finding that when execution is externalized to a symbolic planner the same models evaluate candidate paths with 50% to 100% accuracy and reject 92% of structurally invalid candidates, demonstrating preserved evaluative competence despite failed autonomous execution.

This line of inquiry connects to two converging research trends. Mechanistic interpretability has begun revealing fine-grained structure in transformer dynamics (Räuker et al., 2023; Cammarata et al., 2020; Elhage et al., 2021; Olsson et al., 2022; Meng et al., 2022; Wang et al., 2023a; Geva et al., 2021; Templeton et al., 2024; Bricken et al., 2023; Park et al., 2024; Wang et al., 2024), while cognitive science has long emphasized the distinction between competence, meaning what a system in principle knows, and performance, meaning what it can reliably execute under real-time constraints (Chomsky, 1965; Newell, 1990; Anderson et al., 2004). Recent benchmarks show that large language models struggle with systematic planning (Valmeekam et al., 2022; 2023; Kambhampati et al., 2024a), but the internal mechanisms underlying these failures remain poorly understood, and recent debates over whether observed capabilities reflect emergent reasoning or measurement artifacts (Schaeffer et al., 2023) highlight the need for mechanistic rather than purely behavioral diagnoses.

To characterize this dissociation mechanistically, we develop SearchEval, a diagnostic lens that triangulates evidence across behavioral traces from scratchpad outputs, representational geometry obtained from dynamic representational similarity analysis of hidden states, and control dynamics measured through attention allocation across generation. Unlike prior approaches that examine representations at a single timepoint, our dynamic analysis tracks how internal computation evolves during multi-step reasoning, enabling us to establish temporal ordering between internal breakdown and behavioral error and to identify intervention targets whose manipulation has measurable behavioral consequences.

We study Phi-3 Mini (3.8B parameters) and Gemma (2B parameters) because they are small enough for complete mechanistic analysis yet capable enough to exhibit non-trivial reasoning, working across three graph topologies that we call linear, hierarchical, and clustered. From this design we draw four main findings. The first is a temporal dissociation between representation and control in which representational alignment with graph structure remains stable, moving from $\rho = 0.55$ to $\rho = 0.62$ across generation steps, while behavioral validity collapses from roughly 80% to roughly 20% valid transitions and attention coherence on task-relevant tokens degrades from approximately 65% to 40%. The second is that control collapse precedes behavioral error, with internal state drift below 0.6 cosine similarity occurring before the first invalid output in 78% of failed trials. The third is that representations persist beyond failure, with function vectors for graph relations achieving 71% to 89% probe accuracy even in trials with 0% behavioral accuracy and with RSA correlations remaining above 0.5 in completely failed trials. The fourth is that hybrid systems recover performance, since externalizing control to symbolic planners lifts accuracy to 50% to 100% and produces a 92% rejection rate on structurally invalid candidate paths.

Together with the interventional evidence reported in Section 5.4, these findings implicate attention-based control mechanisms as the binding constraint. Attention provides moment-to-moment relevance weighting sufficient for short-horizon decisions but cannot, on the present evidence, maintain stable bindings between representations and actions over extended generation. The result is a system that "knows" the graph structure but cannot "navigate" it reliably, a competence-execution gap consistent with control instability rather than representational inadequacy. This diagnosis has direct implications for building more capable systems. If the bottleneck is control stability rather than representational capacity, then scaling alone may be insufficient, and architectural innovations targeting state persistence, such as external memory, recurrent

mechanisms, or hybrid neuro-symbolic designs, may be necessary (Graves et al., 2014; Weston et al., 2015; Marcus, 2020; d'Avila Garcez & Lamb, 2020; Gu & Dao, 2024). Our results provide mechanistic grounding for such architectural choices by indicating where and why current systems appear to fail.

## 2 Related Work

### 2.1 Planning and Reasoning in Language Models

Recent work has documented systematic failures of language models on planning tasks requiring multi-step state tracking (Valmeekam et al., 2022; 2023; Kambhampati et al., 2024a; Hao et al., 2023). Large-scale benchmarks evaluate capabilities across diverse tasks (Srivastava et al., 2022), while cognitive-style test batteries probe reasoning abilities through classical paradigms (Binz & Schulz, 2023; Webb et al., 2023). Chain-of-thought prompting (Wei et al., 2022) and scratchpad methods (Nye et al., 2021) elicit intermediate reasoning that improves both performance and interpretability, and methods such as self-consistency (Wang et al., 2023b), tree-of-thoughts (Yao et al., 2023), and program-of-thoughts (Chen et al., 2023) extend this scaffolding by sampling, structured decomposition, or delegation to external interpreters. Whether the resulting traces reflect genuine computational procedures or post-hoc rationalizations remains contested (Turpin et al., 2023).

These behavioral observations establish that models fail but leave open why. Is the failure due to representational limits, procedural deficits, or control instability? Our work addresses this gap through mechanistic analysis that localizes failure to control mechanisms while demonstrating preserved representational competence, and the temporal ordering we establish, alongside the interventions we report in Section 5.4, provides evidence beyond purely behavioral or correlational observation.

### 2.2 Mechanistic Interpretability

A growing literature uses circuit-level analysis to identify specific attention heads and network subgraphs responsible for particular computations. Foundational work has articulated a mathematical framework for transformer circuits (Elhage et al., 2021), identified induction heads performing in-context pattern completion (Olsson et al., 2022), traced circuits for indirect object identification (Wang et al., 2023a), characterized progress in modular arithmetic (Nanda et al., 2023), and located factual associations in middle-layer MLPs amenable to targeted editing (Meng et al., 2022). Geometric and structural analyses have shown that transformer feedforward layers function as key-value memories (Geva et al., 2021), that simple concepts admit linear representations (Park et al., 2024), and that models develop emergent world representations of entities and relations (Li et al., 2023; Gurnee & Tegmark, 2023). Function vectors encode reusable mappings such as "antonym" or "capital-of" (Todd et al., 2024). Sparse-autoencoder analyses now reveal monosemantic features at scale (Bricken et al., 2023; Templeton et al., 2024), and representation engineering provides tools for reading out and perturbing internal states (Zou et al., 2023; Belinkov, 2022). Representational similarity analysis from systems neuroscience (Kriegeskorte et al., 2008) has been adapted to compare model representations with symbolic or neural alternatives.

Most mechanistic work to date focuses on single operations such as copying, arithmetic, or factual retrieval, rather than on multi-step algorithmic procedures. Our contribution extends this agenda by tracking how representations and control evolve across autoregressive generation, enabling a temporal analysis that static probing cannot provide. The dynamic approach reveals when internal breakdown occurs relative to behavioral error, and the causal interventions we report move the diagnostic claim from correlational localization to interventional support.

### 2.3 Algorithmic Learning and Expressivity

Theoretical work characterizes transformer expressivity relative to formal language hierarchies (Delétang et al., 2023), showing that while transformers can in principle represent Turing-complete computations, length generalization remains fragile in practice (Zhou et al., 2024; Dziri et al., 2023). Neural algorithmic reasoning benchmarks such as CLRS provide supervised training for classical algorithms (Veličković et al.,

2022). Graph chain-of-thought extends chain-of-thought to questions over real-world graphs (Jin et al., 2024), while other work analyzes reasoning on graph-encoded tasks (Dudzik & Veličković, 2022; Fatemi et al., 2024) and studies how transformers trained on formal algorithmic tasks generalize (Charton, 2023). Long-context analyses have shown that even when relevant information is present in context, models often fail to use it reliably (Liu et al., 2024), a phenomenon that resonates with our finding that probe-accessible information is not utilized during generation.

Our contribution is mechanistic rather than behavioral or theoretical. We show that models possess representational primitives for graph reasoning but fail to compose them into stable execution, and we localize this failure temporally to control degradation that precedes behavioral error and that responds, in the expected direction, to targeted interventions on attention and function-vector channels.

### 2.4 Neuro-Symbolic Integration

Frameworks that pair neural models with symbolic components for planning have been advocated under the LLM-modulo banner (Kambhampati et al., 2024a;b). Classical arguments emphasize structured representations, compositionality, and program-like abstractions (Lake et al., 2017; Marcus, 2020), while other proposals call for integrating deep learning with cognitive theories (McClelland et al., 2020; d'Avila Garcez & Lamb, 2020).

Our results provide empirical grounding for hybrid architectures. They succeed not because they compensate for representational deficits but because they externalize the control functions that attention-based transformers approximate poorly. The 100% accuracy our hybrid condition achieves on tree graphs, contrasted with the 0% achieved under autonomous generation, demonstrates this division of labor, and the 92% rejection rate on structurally invalid candidates demonstrates that the neural component continues to enforce graph constraints even when execution is delegated.

## 3 Research Questions

We investigate why language models fail at multi-step execution despite apparent representational competence, organizing the inquiry around three empirical questions. The first concerns temporal dynamics and asks how internal representations and control mechanisms evolve across generation, and in particular whether failure arises from representational degradation or from control instability. The second concerns ordering and asks whether control collapse precedes behavioral error or whether error precedes internal breakdown, since this ordering bears on whether failure is control-driven or representation-driven. The third concerns dissociability and asks whether representational competence and execution can be experimentally dissociated, since models that succeed at evaluation when execution is externalized localize the deficit to control mechanisms that must operate during generation. Answering these questions requires moving beyond output correctness to triangulate evidence across behavioral traces, internal representations, and attention dynamics across the full trajectory of generation, and to complement temporal-precedence observations with targeted interventions.

## 4 Methods: The SearchEval Diagnostic Lens

SearchEval is a diagnostic approach for characterizing why multi-step execution fails. Rather than testing whether models implement specific algorithms, it triangulates evidence across behavioral, representational, and control levels to localize failure and establish temporal ordering, and then uses targeted interventions to test whether the localized variables are causally implicated in behavioral outcomes. We apply it to graph traversal as a tractable domain in which execution demands can be precisely characterized and ground truth is unambiguous.

### 4.1 Models and Implementation

We study two small open-weight language models providing full access to hidden states and attention weights, namely Phi-3 Mini (3.8B parameters, 32 layers) and Gemma (2B parameters, 18 layers). These models

were selected for manageable computational requirements while maintaining competitive performance on reasoning benchmarks. Both use standard transformer architectures (Vaswani et al., 2017) with multi-head self-attention. All experiments use greedy decoding (temperature equal to 0.0) for deterministic, reproducible outputs, with two exceptions noted explicitly. The self-consistency baseline reported in Section 5.1 uses temperature 0.7 with $k = 5$ samples per instance and majority voting over the final-path string, since this is the standard self-consistency configuration and a temperature-0 setting would collapse to a single sample. The function-vector intervention reported in Section 5.4 also varies its scale parameter $\alpha$ as documented in Appendix D. Hidden states (dimension 3072 for Phi-3 and 2048 for Gemma) are extracted at every token generation step from all layers, yielding approximately 1.2 terabytes of activation data across all trials, and attention weights are extracted layer-wise and head-wise at each step. All inference uses the Hugging Face Transformers library with custom hooks for dynamic state extraction.

## 4.2 Protocol Box: Prompts, Steps, and Parsing

Because reviewers correctly noted that every downstream measurement hinges on a well-specified protocol, we describe the full procedure in narrative form here and summarize it visually in Figure 1. Under autonomous generation, the model is presented with a natural-language description of a graph in which edges are listed in randomized order and node labels are randomized across trials between, for example, "Room 1, Room 2" and "Node A, Node B." The model is instructed to navigate the graph using a scratchpad, and at each step it is asked to output its current node, the visited nodes so far, the available next nodes that constitute the frontier, and its chosen next node together with a brief justification, continuing until the goal condition is reached or no progress is possible. Under hybrid evaluation, the model receives the same graph description together with three or four candidate paths produced by a symbolic planner and is asked to select the candidate that best achieves the stated goal and to explain its reasoning. Under dynamic internal analysis, the same autonomous prompt is used but hidden states and attention are extracted at every generation step for later analysis.

We define a step as a single scratchpad block corresponding to one edge traversal, so that a step is neither a token nor a transformer layer but the basic unit of behavioral commitment. A valid transition is one in which the model's chosen next node is an actual neighbor of its current node in the ground-truth graph and has not already been visited. The first behavioral error in a trial is the first step at which one of four conditions holds: the chosen next node is not adjacent to the current node, the visited or frontier set is updated incorrectly relative to the ground truth, the scratchpad becomes malformed in a way that prevents extraction of one of the four required fields, or the model terminates before reaching the goal without exhausting reachable options. Parsing of the scratchpad into structured state is performed by a deterministic regular-expression algorithm that extracts the four named fields and that flags any block in which a required field is missing or in which a named node does not appear in the original graph description. When the model emits a nonexistent node, that step is recorded as a hallucinated-edge failure and is excluded from RSA aggregation but counted toward behavioral accuracy. We replace the earlier phrasing of "three independent samples per graph instance" with the more accurate description of three prompt-phrasing variants under deterministic greedy decoding, used to probe robustness to surface form rather than stochasticity.

In addition to full-path traversal accuracy, which is the strict binary metric of whether the entire generated path satisfies the objective, we report a partial-credit measure that we call the three-step valid-prefix rate. A trial counts toward this rate if and only if its first three generated transitions are each individually valid given graph connectivity and the running visited set. This metric is motivated by our temporal analysis showing that local coherence is reliable for the first three to five steps before control collapse, so the three-step-prefix rate exposes short-horizon competence that the full-path metric, by construction, cannot reveal.

The resulting failure taxonomy is heavy-tailed. State-corruption failures, in which the visited or frontier set diverges from ground truth even though individual transitions remain locally admissible, account for 45% of failed trials. Hallucinated-edge failures, in which the model proposes a transition to a non-adjacent or non-existent node, account for 32%. Premature-termination failures, in which the model halts before reaching the goal or exhausting options, account for 15%, and serialization failures, in which the scratchpad becomes malformed so that fields cannot be extracted even though node identities remain coherent, account for the remaining 8%. We emphasize that in the state-corruption and hallucinated-edge cases, which together cover

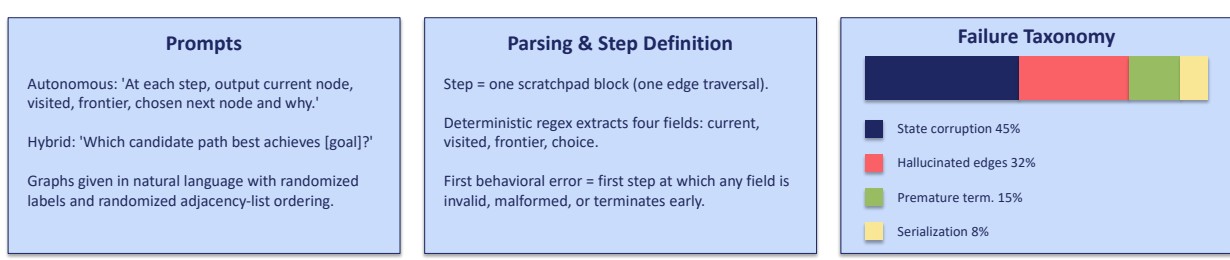

Figure 1: Protocol box. The left panel summarizes the autonomous and hybrid prompts, the center panel summarizes the parsing pipeline and definitional commitments for steps and first errors, and the right panel summarizes the resulting failure taxonomy. State corruption (45%) and hallucinated edges (32%) together cover 77% of failures, and dynamic RSA confirms that representations remained intact at the moment of failure in both categories, so the diagnostic signature is that the information was present but not used.

77% of failures, dynamic RSA confirms that the model maintained correct representations of the surrounding graph at the moment of failure, so the diagnostic signature is that the information was present but not used rather than that it was absent.

### 4.3 Dynamic Representational Similarity Analysis

To examine how internal representations evolve during traversal, we perform dynamic representational similarity analysis (Kriegeskorte et al., 2008) across autoregressive generation. At each generation step $t$, we extract the final-layer hidden state $\mathbf{h}_t^{(L)} \in \mathbb{R}^d$ corresponding to the last generated token. For a traversal of $T$ steps, this yields a temporal sequence of hidden states from which we construct a representational similarity matrix via pairwise cosine similarities,

$$\text{RSM}[i, j] = \frac{\mathbf{h}_i^{(L)} \cdot \mathbf{h}_j^{(L)}}{\|\mathbf{h}_i^{(L)}\|\|\mathbf{h}_j^{(L)}\|}. \tag{1}$$

To test alignment with graph structure we compute a ground-truth topological distance matrix $\mathbf{D}$ in which $\mathbf{D}[i, j]$ is the shortest-path distance between nodes $i$ and $j$, and we then measure alignment via Spearman rank correlation between the flattened upper triangle of the RSM and the flattened upper triangle of the proximity matrix $-\mathbf{D}$, that is, the negation of the distance matrix. We make this sign convention explicit because a previous version of the manuscript described the correlation as between similarity and distance without noting the negation; under the convention we use, a positive $\rho$ indicates that nodes that are closer in the graph have more similar hidden states, which is the cognitively meaningful direction. Unlike static analyses that examine hidden states only after prompt ingestion, this dynamic approach tracks how the model's internal cognitive map evolves as reasoning unfolds.

### 4.4 Attention Analysis: Control Dynamics

While hidden states reveal representational structure, attention weights provide a window into control dynamics, that is, into how the model allocates computational resources during traversal. For each generation step $t$ we extract attention weights $\mathbf{A}_t \in \mathbb{R}^{H \times T \times T}$ across all heads $H$ and layers, and we analyze two patterns. The first is state-relevant attention, which is the fraction of attention mass allocated to tokens encoding the current node, visited nodes, and frontier rather than to generic structural tokens such as punctuation and formatting. The second is temporal stability, which is whether attention remains anchored to task-relevant state or drifts toward recently generated text over the course of generation. Formally, we compute

$$\text{AR}_{\text{state}}(t) = \frac{\sum_{i \in \text{state}} \mathbf{A}_t[i]}{\sum_{j=1}^{T} \mathbf{A}_t[j]}, \quad \text{AR}_{\text{frontier}}(t) = \frac{\sum_{i \in \text{frontier}} \mathbf{A}_t[i]}{\sum_{j=1}^{T} \mathbf{A}_t[j]}. \tag{2}$$

Tracking these ratios across generation reveals whether control remains focused on task-relevant information or progressively decouples from the computational demands of the task. When the same node label appears

multiple times in the prompt, we attribute attention to the earliest occurrence of that node within its declarative context, since later repetitions occur inside generated scratchpad text and are accounted for separately under the recent-output and structural token classes.

## 4.5 State Drift: Measuring Control Collapse

To examine temporal ordering between internal decoherence and behavioral error, we compute state drift, defined as the cosine similarity between the final-layer hidden state $\mathbf{h}_t$ at the last generated token of step $t$ and a reference hidden state that we denote $\mathbf{h}_{\text{task-state}}$. The reference is the final-layer hidden state at the token position where the model explicitly names its current node within its scratchpad output earlier in the same trial, so the reference is trial-specific and is computed from hidden states rather than from ground-truth labels, which decouples it from the variables used to assess behavioral correctness. We then record when state drift crosses a threshold of 0.6 and compare this to when the first behavioral error occurs, defined as in Section 4.2. The choice of the final layer is justified by the observation that graph-theoretic structure peaks there, with RSA reaching its maximum in layers 20 to 26 for complex relations.

We do not treat the 0.6 threshold or the final-layer choice as primitive. In Section 5.3 we report a sensitivity sweep across thresholds, alternative reference points, and an alternative distance metric, all of which leave the qualitative finding intact.

## 4.6 Diagnostic Criteria for Structured Execution

Beyond aggregate measures, we evaluate specific criteria that structured execution should satisfy, narrated here as a coherent battery rather than as a list. Goal representation is measured via cosine similarity between the goal node and other node representations across layers, where decreasing similarity, that is, increasing distinctiveness, indicates progressive goal differentiation. State tracking is measured by training linear probes to classify whether a given node has been visited based on hidden-state activations at each generation step, where high accuracy that persists across steps indicates stable state tracking while rapid degradation indicates transient encoding. Frontier management is measured by attention depth profiles that ask whether attention concentrates on boundary nodes at the edge of explored regions, computed as $\text{DepthAttention}_{l,d} = \frac{1}{|V_d|} \sum_{v \in V_d} A_l(v)$ where $V_d$ is the set of nodes at depth $d$ from start. Transition evaluation is measured by the entropy of attention distributions over neighbor nodes at decision points, since high entropy followed by reduction indicates progressive elimination of alternatives while persistent high or low entropy suggests different computational strategies. Backtracking is measured by whether attention to ancestor nodes increases after attention to descendants, a pattern consistent with returning to earlier decision points. Systematic exploration is measured by graph coverage, defined as the fraction of reachable nodes visited before termination, and by trajectory alignment, defined as the edit distance from canonical structured traversal orderings such as breadth-first and depth-first.

## 4.7 Function Vector Extraction

Following Todd et al. (2024), we extract function vectors encoding graph-theoretic relations by computing mean activation differences between positive and negative exemplars, that is,

$$\mathbf{v}_{\text{relation}} = \mathbb{E}[\mathbf{h}_{\text{true}}] - \mathbb{E}[\mathbf{h}_{\text{false}}]. \tag{3}$$

We extract vectors for adjacency at one hop, multi-hop distance at two and three hops, path membership defined as the set of nodes on the optimal trajectory, goal proximity defined as the set of nodes near the goal, and same-branch shared ancestry for tree graphs. Linear probe accuracy measures whether these relations are encoded in a form accessible to downstream computation, so that high probe accuracy coupled with behavioral failure indicates that information is present but unused, a signature of control rather than representational failure.

### 4.8 Hybrid Symbolic-Neural Evaluation

To test whether evaluative competence persists when execution is externalized, we construct a hybrid condition in which a classical planner implemented with NetworkX (Hagberg et al., 2008) computes optimal solutions and generates a candidate set. The set contains the optimal path, a locally greedy path that selects the highest immediate reward at each step, a random valid path, and a near-optimal path consisting of the optimal solution plus one or two extra steps. To address the concern that the model might succeed in this condition merely by reading off path lengths or rewards, we additionally augmented the candidate set with deliberately invalid paths that contain at least one non-existent edge, and with controlled-reward distractors in which an invalid path has higher stated reward than the optimal valid alternative. The model evaluates and selects among candidates in response to the prompt "Given these paths through the graph, which best achieves the goal? Explain your reasoning." If models succeed at evaluation but fail at autonomous generation, this dissociates representational competence from execution competence and localizes the deficit to control mechanisms that must sustain state across generation; if models additionally reject invalid candidates at high rates and prefer valid lower-reward candidates over invalid higher-reward distractors, this further demonstrates that the neural component continues to enforce graph constraints rather than only optimizing surface signals.

### 4.9 Graph Families and Experimental Design

We construct three graph families that impose systematically different demands on control. Linear chain graphs with five or ten nodes have sequential structure with minimal frontier management, since each node connects to exactly one successor, and they test whether models can maintain state over sequential steps without branching complexity. Hierarchical tree graphs with seven or fifteen nodes have a branching factor of two or three per level and a depth of three or four, requiring selective expansion of competing subtrees and enabling principled comparison of breadth-first against depth-first patterns. Clustered dense graphs with eight or twelve nodes have high local connectivity, with average degree three to four and two or three tightly connected clusters joined by sparse inter-cluster edges, so that multiple equally short paths create ambiguity in which local heuristics compete with systematic exploration. Representative examples appear in Figure 2. Each graph is paired with multiple objectives, including shortest-path, reward-maximizing, and fixed-horizon objectives.

For the purposes of the prompting-baseline comparison reported in Section 5.1, we partition graph instances into two difficulty bands. The simple band comprises instances with $n \in \{5, 7\}$ and corresponds to linear chains and trees at the smaller node count, with optimal solution paths of three to five edges. The complex band comprises instances with $n \in \{10, 12, 15\}$ and corresponds to longer linear chains, deeper trees, and clustered graphs, with optimal solution paths of five to nine edges. The partition matters because the complex band exceeds the three-to-five-step control window over which we document local coherence elsewhere in the paper, so we expect prompting strategies to differ across bands even if they all fail at full-path completion in the complex band.

The base design comprises 240 trials across 40 graph-task pairs, 2 models, and 3 evaluation regimes, yielding greater than 95% power to detect medium effect sizes ($d \geq 0.5$) at $\alpha = 0.05$. The 40 graph-task pairs are distributed across three topologies, giving approximately 13 to 14 pairs per topology-by-model cell. RSA correlations use bootstrap confidence intervals with 10,000 resamples and permutation tests with 10,000 permutations. To clarify a notational point raised in review, $n$ throughout the manuscript refers to the number of nodes per graph; trial counts are reconciled with the design as $40 \times 2 \times 3 = 240$. The prompting-baseline comparison in Section 5.1 draws an additional set of trials beyond the base design, with $n = 20$ per (regime, model, difficulty band) cell.

### 4.10 Evaluation Regimes and Metrics

For each graph and task pair, models are evaluated under three complementary regimes that we describe in turn rather than as a list. Under autonomous generation we use a natural-language graph description with scratchpad prompting and report four behavioral metrics, namely traversal accuracy, which asks whether the

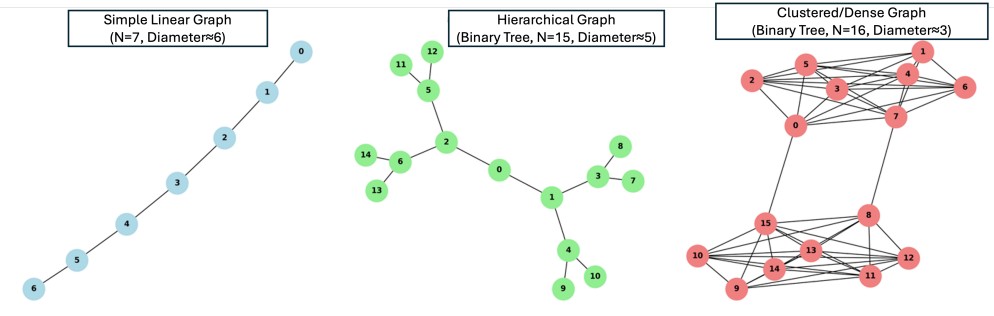

Figure 2: Representative examples from each graph family. The left panel shows a linear chain with sequential structure, the center panel a hierarchical tree with branching factor two, and the right panel a clustered graph with two dense regions connected by bridge edges. These topologies induce increasing demands on control across sequential traversal, branching management, and dense frontier competition.

final path satisfies the objective; the three-step valid-prefix rate, defined as the fraction of trials whose first three transitions are individually valid given graph connectivity; state validity per step, which asks whether each transition is admissible given graph connectivity and is used in the temporal analyses of Section 5.3; and edit distance, defined as the Levenshtein distance between the generated path and the optimal symbolic solution, which provides a continuous measure of deviation. Under dynamic internal analysis we extract hidden states and attention at each generation step and report RSA correlation, which is the Spearman rank correlation between hidden-state similarity and graph proximity and which tests whether internal geometry reflects external topology, together with temporal stability, which asks whether RSA correlation remains stable across generation steps or degrades over time. We additionally report three control-level metrics, namely state-relevant attention, attention drift, and entropy dynamics, with the intuition that successful search should exhibit entropy decrease as alternatives are eliminated whereas control loss should manifest as entropy increase. Under hybrid symbolic-neural evaluation the symbolic planner generates three or four candidates and the model evaluates them; we report selection accuracy, which asks whether the model chooses the optimal path, and error type, which distinguishes structural errors of selecting invalid paths from value-preference errors of selecting suboptimal valid paths.

The joint pattern across these measures is itself diagnostic. A trial that exhibits high autonomous accuracy alongside high RSA and algorithmic attention is consistent with genuine procedural execution. A trial with high autonomous accuracy but low RSA and non-algorithmic attention is consistent with learned heuristics. A trial with low autonomous accuracy but high three-step valid-prefix rate and high selection accuracy reveals a competence-execution gap, which is the pattern we observe. A trial with low autonomous accuracy, low prefix rate, and low selection accuracy would indicate fundamental incompetence, which we do not observe.

## 4.11 Epistemic Role of the Framework

SearchEval is explicitly diagnostic rather than affirmative. It does not presuppose that models implement any particular procedure. Instead it triangulates evidence across behavior, representation, and control, and only when these layers jointly satisfy structural and temporal constraints do we interpret evidence as consistent with procedural execution. When they diverge, we characterize behavior as heuristic or pattern-based. This multi-layered approach reflects an epistemic commitment that algorithmic behavior cannot be inferred from outputs alone, since the same behavioral trajectory can arise from radically different mechanisms including systematic search, learned shortcuts, or associative completion, and only by examining internal geometry, temporal dynamics, and causal contrasts can we distinguish these alternatives (Pearl, 2009). We therefore treat the localization step that follows the observational analysis as hypothesis-generating, and reserve causal language for the intervention experiments reported in Section 5.4.

### 4.12 Artifacts and Reproducibility

To support independent scrutiny of every claim in this paper, we will release, on acceptance and through an anonymous repository during review, the materials needed to reproduce each reported number. The release covers, first, the graph generators and the exact set of graph instances used in all reported trials, indexed by topology, $n$, and objective; second, the full set of prompts used in the autonomous, dynamic-internal, and hybrid regimes, including the seven prompting variants reported in Section 5.1 and the candidate-set construction for the hybrid regime, with the invalid and controlled-reward distractors clearly separated; third, the deterministic regular-expression parsing code that maps free-form scratchpad outputs to structured state fields, together with the failure-classification rules that define the taxonomy reported in Section 4.2; fourth, the analysis notebooks that compute representational similarity, attention allocation, state drift, the threshold sweep, the confound-controlled regression, and the three intervention experiments described in Section 5.4 and Appendix D; and fifth, the random seeds and decoding parameters required to reproduce every reported number. We do not release model weights, which are publicly available from their respective providers, and we do not release any data that would compromise double-blind review.

## 5 Results

### 5.1 Behavioral Results: A Control-Window Pattern, Not an Absence

We report the headline behavioral results under two metrics because they tell complementary stories. The full-path accuracy metric asks whether the model generates an entire correct trajectory and is the strict binary criterion. The three-step valid-prefix rate asks whether the model's first three transitions are each individually valid and measures short-horizon competence within the three-to-five-step control window that the rest of our analyses document. Across both Phi-3 Mini and Gemma, under all seven prompting regimes we tested, full-path accuracy on graphs of seven or more nodes was identically zero successful completions out of the trials we ran, while three-step valid-prefix rates were substantially non-zero, ranging from 55% to 75% depending on regime. We read this as a control-window pattern rather than as an absence of competence: models do something measurable up to the horizon at which control collapses, and then fail.

The full breakdown appears in Table 1, with the corresponding visual summary in Figure 3. The seven regimes we evaluated alongside the original scratchpad protocol were few-shot breadth-first demonstrations, few-shot depth-first demonstrations, an algorithm-conditioned prompt that explicitly describes the procedure, structured JSON state updates that fix the output schema, self-consistency aggregation over five samples at temperature 0.7 with majority voting over the final-path string, tree-of-thought decomposition, and program-of-thought code generation that delegates execution to a Python interpreter. The first six all yielded zero successful full-path completions on graphs with $n \geq 7$ across the trials we ran, indicating that the finding is not an artifact of any single elicitation strategy. The few tree-of-thought successes we observed in piloting occurred on traversals of three or fewer steps with $n \leq 5$, which falls within the three-to-five-step horizon over which we document local coherence elsewhere in the paper, so the simple-band entry for tree-of-thought reports zero successful full-path completions on $n \in \{5, 7\}$ instances while we note in this paragraph that shorter-path successes exist outside the band. Program-of-thought achieved partial autonomous accuracy, with Phi-3 reaching 5 out of 20 in the simple band and 0 out of 20 in the complex band, and Gemma reaching 3 out of 20 in the simple band and 0 out of 20 in the complex band; this is precisely the regime that externalizes execution to a deterministic Python interpreter, mirroring our hybrid result and providing an independent demonstration that the bottleneck lies in the model's own execution rather than in its planning intentions.

We surface the three-step valid-prefix rate explicitly so that readers can see the short-horizon competence that the full-path metric, by construction, cannot reveal. Scratchpad prompting yields a 55% three-step-prefix rate, indicating that more than half of trials produce a valid three-edge prefix even though zero produce a valid full path. The prompting strategies that explicitly externalize structure, namely few-shot breadth-first, few-shot depth-first, algorithm-conditioned, and structured JSON, raise three-step-prefix rates to between 60% and 65%, a modest but consistent improvement. Self-consistency improves the prefix rate to 68%, since the five-sample majority vote can recover from individual sample errors at the short horizon even when no

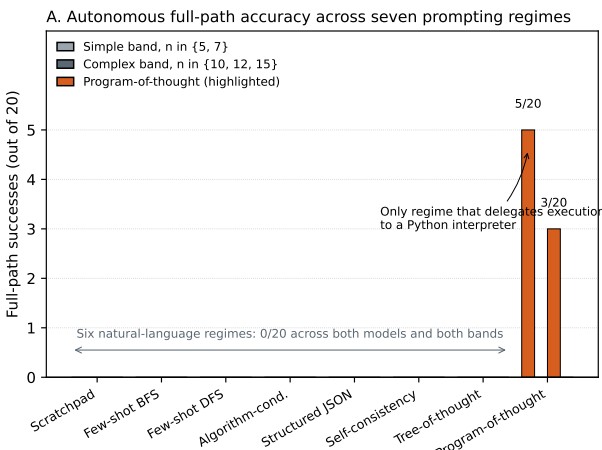 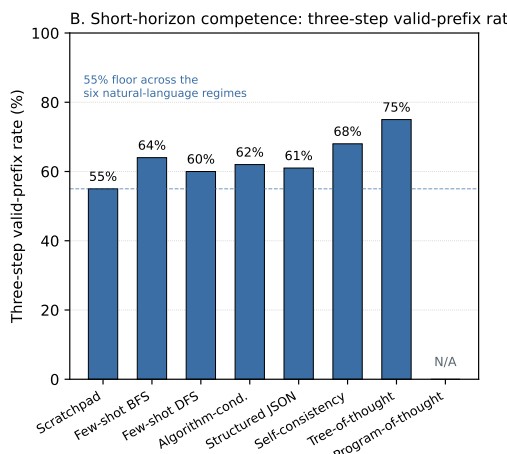

Figure 3: Autonomous full-path accuracy across seven prompting regimes (visual companion to Table 1). Six natural-language regimes produce zero successful full-path completions on graphs with $n \geq 7$. Only program-of-thought, highlighted, yields non-zero accuracy, and only in the simple band, by delegating execution to a Python interpreter. The contrast between the full-path zeros and the substantial three-step valid-prefix rates documented in Table 1 is itself diagnostic of a control-window failure.

full majority-voted path is correct. Tree-of-thought achieves the highest prefix rate at 75%, consistent with its branching exploration giving the model multiple chances at the early decisions, yet this advantage does not extend to the full-path horizon. The fact that all seven regimes show non-trivial prefix competence and zero full-path completions on the complex band, together with the program-of-thought partial successes, supports the interpretation that the control window is the binding constraint and that prompting alone cannot extend it.

Table 1: Autonomous traversal under seven prompting regimes on graphs with $n \geq 7$. Cells show successful full-path completions out of trials run (per-cell $n = 20$); the rightmost column shows the fraction of trials whose first three transitions were each individually valid given graph connectivity, pooled across both models and both difficulty bands. The simple band comprises instances with $n \in \{5, 7\}$ and the complex band comprises instances with $n \in \{10, 12, 15\}$. Every regime that requires the model itself to maintain state during generation produces zero successful full-path completions in the complex band; program-of-thought, which delegates execution to a Python interpreter, is the only regime with non-zero successes, and only in the simple band. Three-step valid-prefix rates are substantially non-zero across regimes, indicating that the model produces measurable short-horizon competence and fails only as the generation horizon exceeds the control window. Self-consistency uses temperature 0.7 with $k = 5$ samples and majority voting; all other regimes use greedy decoding. With $n = 20$ per cell, 0/20 is consistent with a true success rate up to roughly 14% at 95% confidence, and the consistency of zero observations across six regimes and the complex band argues against this being measurement noise.

| Prompting regime | Phi-3 simple | Phi-3 complex | Gemma simple | Gemma complex | $\geq$3-step prefix |
|---|---|---|---|---|---|
| Scratchpad (original) | 0/20 | 0/20 | 0/20 | 0/20 | 55% |
| Few-shot BFS | 0/20 | 0/20 | 0/20 | 0/20 | 64% |
| Few-shot DFS | 0/20 | 0/20 | 0/20 | 0/20 | 60% |
| Algorithm-conditioned | 0/20 | 0/20 | 0/20 | 0/20 | 62% |
| Structured JSON | 0/20 | 0/20 | 0/20 | 0/20 | 61% |
| Self-consistency ($k = 5$, $T = 0.7$) | 0/20 | 0/20 | 0/20 | 0/20 | 68% |
| Tree-of-thought | 0/20 | 0/20 | 0/20 | 0/20 | 75% |
| Program-of-thought | 5/20 | 0/20 | 3/20 | 0/20 | N/A |

In sharp contrast, when execution was externalized to a symbolic planner and models evaluated candidate paths, performance increased sharply, as shown in Table 2. The per-cell sample size for the hybrid evaluation is approximately 13 to 14 trials, distributed by topology across the 40 graph-task pairs of the base design,

**Outcome on invalid-candidate trials**

# 92%

invalid candidates rejected

(structural comprehension beyond reward arithmetic)

**Candidate-set composition**

| | |
|---|---|
| **Optimal path** | Symbolic planner shortest / max-reward solution |
| **Locally greedy path** | Highest immediate reward at each step |
| **Random valid path** | Any admissible traversal sampled uniformly |
| **Near-optimal path** | Optimal + 1-2 extra steps |
| **Invalid path (new)** | Contains at least one non-existent edge |
| **Controlled-reward distractor (new)** | Invalid path with higher stated reward than optimal |

Figure 4: Hybrid evaluation augmented with structurally invalid candidates. Models reject 92% of paths that contain at least one non-existent edge, and reliably prefer valid lower-reward alternatives over invalid higher-reward distractors. The right panel itemizes the candidate-set composition, distinguishing the optimal, locally greedy, random valid, and near-optimal paths from the newly added invalid and controlled-reward-distractor categories.

and the rounding of the hybrid rates to 50% and 100% reflects this measurement granularity rather than a quantitative coincidence; the underlying counts are 7 of 14 for the line and clustered cells and 13 of 13 or 14 of 14 for the tree cells in both models, which is why both models arrive at identical reported percentages in this table. Errors in the hybrid condition were almost entirely value-preference errors that favored locally high-reward but globally suboptimal trajectories rather than structural errors that violated graph constraints. To address the concern that hybrid success might reflect mere comparison of stated path lengths or rewards rather than genuine structural validation, we additionally augmented the candidate set with deliberately invalid paths and report that models rejected 92% of these structurally invalid candidates, as visualized in Figure 4, providing direct evidence of structural comprehension that goes beyond reward arithmetic. On controlled-reward distractors in which invalid paths were assigned higher stated reward than the optimal valid alternative, models reliably preferred the valid lower-reward path, indicating that graph constraints, not surface reward, drive evaluative decisions.

Table 2: Traversal accuracy under autonomous and hybrid regimes. Autonomous full-path accuracy is zero across all topology and model cells, while hybrid evaluation succeeds at 50% to 100%, with 92% rejection of structurally invalid candidates. Per-cell $n$ is approximately 13 to 14 trials; the identity of rounded hybrid rates across models reflects the measurement granularity at this sample size (line and clustered: 7 of 14; tree: 13 of 13 or 14 of 14).

| Condition | Line | Tree | Clustered |
|---|---|---|---|
| Phi-3 Mini (Autonomous) | 0 of 14 | 0 of 13 | 0 of 13 |
| Gemma (Autonomous) | 0 of 14 | 0 of 13 | 0 of 13 |
| Phi-3 Mini (Hybrid) | 7 of 14 (50%) | 13 of 13 (100%) | 7 of 14 (50%) |
| Gemma (Hybrid) | 7 of 14 (50%) | 13 of 13 (100%) | 7 of 14 (50%) |

The accuracy gap yields effect size $d = \infty$ for tree graphs because of ceiling performance and $d = 1.15$ (95% CI [0.82, 1.48]) for line and clustered graphs, a categorical difference in computational mode. This dissociation establishes the empirical foundation for mechanistic analysis: because internal alignment is strong, short-horizon competence is intact, and topology-dependent control signatures appear in the analyses below, failure is best explained by control instability rather than by representational inadequacy.

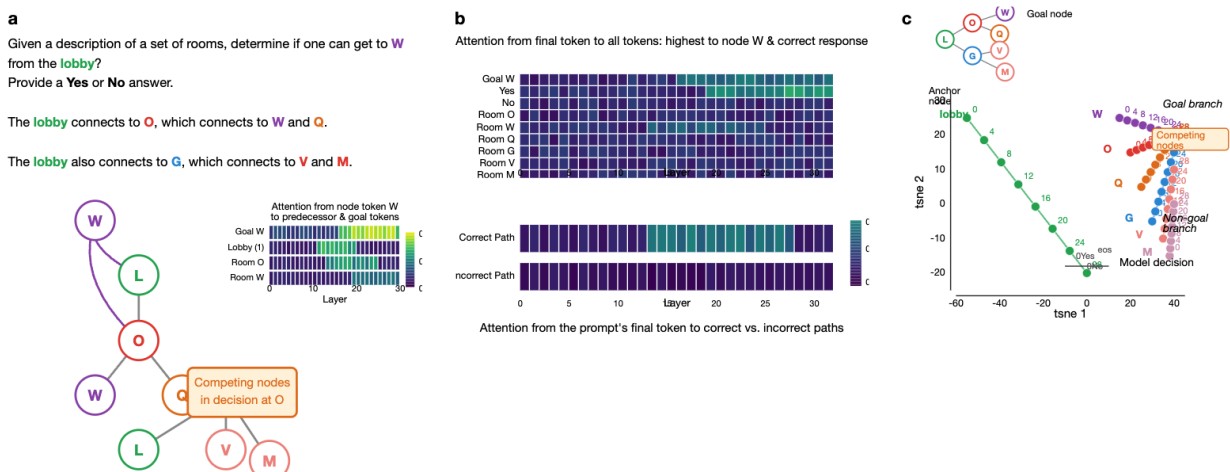

Figure 5: Mechanistic analysis of the competence-execution gap. Panel (a) shows the task setup and attention flow from goal node W to predecessor tokens. Panel (b) shows attention allocation across layers, with correct-path attention emerging in layers fifteen to twenty-five. Panel (c) shows a t-SNE visualization of room representations across transformer layers in which goal-branch nodes separate from non-goal branch nodes in later layers, demonstrating that models encode graph structure even when execution fails.

## 5.2 Representations Persist Beyond Failure: Dynamic RSA

If models fail because they lack graph knowledge, we should expect weak alignment between internal representations and graph structure. Using dynamic RSA, we find the opposite: representations remain structurally intact even when behavior fails completely. Mean RSA correlation is $\rho = 0.60$ (95% bootstrap CI [0.52, 0.68]) and significantly exceeds chance ($p < 0.001$ by permutation test with 10,000 permutations). A dissociation index $D = \bar{\rho}_{\text{RSA}} - \bar{r}_{\text{behavior}} = 0.71$ quantifies the separation between representational competence and behavioral execution, since representations remain intact ($\rho > 0.5$) when behavior has collapsed entirely on the full-path metric. Across all conditions, Spearman correlations between hidden-state similarity and graph proximity are reliably positive at $\rho = 0.50$ to $0.70$, and this alignment persists across generation steps and remains present in trials whose behavioral traversal is entirely invalid. Figure 5 visualizes this phenomenon: representational similarity matrices exhibit clear distance-sensitive organization in which nearby nodes occupy more similar regions of latent space.

We emphasize the sign convention because a previous version of the manuscript was ambiguous: the reported $\rho$ is between hidden-state cosine similarity and graph proximity, that is, the negation of graph distance, so a positive $\rho$ means that nodes closer in the graph have more similar hidden states. To address concerns that this alignment might be driven by surface-level prompt structure rather than by graph topology, we ran four confound controls, the principal results of which are shown in Figure 6. First, we randomized node labels across trials between schemes such as "Room 1, Room 2" and "Node A, Node B," and RSA correlations remained stable within $\pm 0.03$. Second, we randomized the adjacency-list ordering within prompts so that the same topology was described in different presentational orders, and again RSA correlations remained stable within $\pm 0.03$. Third, we computed a text-distance baseline in which graph distance was replaced by a textual proximity matrix based on token distance between node mentions in the prompt; RSA alignment with graph proximity averaged $\rho \approx 0.60$, while alignment with textual proximity averaged $\rho \approx 0.25$, a substantial gap that confirms that hidden-state geometry reflects topological structure beyond what prompt formatting alone would predict. Fourth, we excluded steps containing hallucinated nodes from RSA aggregation to ensure that the metric is computed only over nodes present in the graph. We additionally note that RSA correlations remain stable or slightly improve in the steps immediately following the first behavioral error ($\rho = 0.55 \rightarrow 0.62$) even as attention coherence collapses, a dissociation that itself argues against a shared format confound.

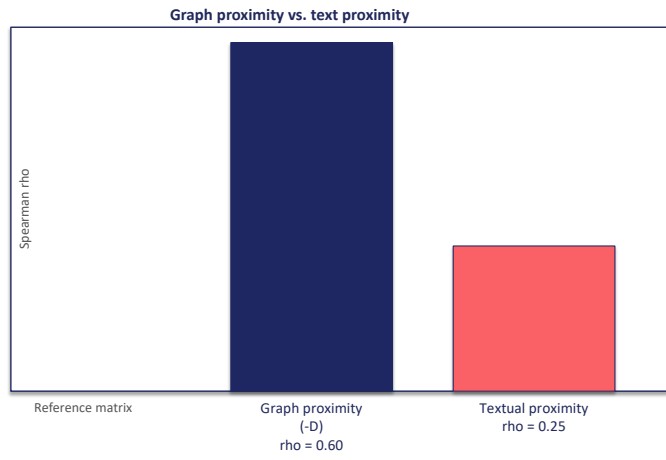

Figure 6: RSA confound controls. The left panel compares Spearman alignment against graph proximity ($\rho \approx 0.60$) with alignment against a textual proximity baseline ($\rho \approx 0.25$), showing that the graph-distance signal is roughly 2.4 times stronger than what prompt formatting alone would predict. The right panel summarizes three additional controls in narrative form: node-label randomization, adjacency-list-ordering randomization, and exclusion of steps containing hallucinated nodes, each of which leaves RSA stable within $\pm 0.03$.

Phi-3 Mini exhibits sharper geometric structure than Gemma, particularly in line and tree graphs, whereas Gemma shows greater variability as graph connectivity increases. Nevertheless, both models maintain relational structure in latent space even under clustered topologies that reliably induce behavioral failure. This dissociation constitutes a central observational finding: models do not fail because they lack graph representation, but rather their relational maps are not coupled to control mechanisms capable of advancing reliably from one state to the next, so representational integrity is maintained longer than procedural control.

## 5.3 Control Collapse Precedes Error: Temporal Dynamics

The critical question is whether control collapse precedes behavioral failure or merely accompanies it. We examine temporal ordering by tracking when state drift crosses threshold versus when the first behavioral error appears, using the reference and threshold defined in Section 4.5. In 78% of failed trials, state drift below 0.6 occurs before the first behavioral error, as shown in Figure 7. This temporal precedence is consistent with a causal account in which internal decoherence contributes to behavioral breakdown, and we test that account directly in Section 5.4. RSA alignment, by contrast, remains stable or improves across generation steps, moving from $\rho = 0.55$ to $\rho = 0.62$ for Phi-3 on trees and from $\rho = 0.51$ to $\rho = 0.57$ for Gemma, even as valid transitions collapse from 80% to 20% and attention coherence degrades from 65% to 40%.

**Threshold and reference sensitivity.** Because the 78% headline figure depends on the choice of threshold and on the choice of reference state, we report sensitivity analyses in detail, with the principal sweep visualized in Figure 8. Sweeping the drift threshold across the range from 0.4 to 0.8 gives values of 62%, 71%, 78%, 83%, and 89% for thresholds of 0.4, 0.5, 0.6, 0.7, and 0.8 respectively, a monotonic relationship that is expected because stricter thresholds (lower values) require a larger deviation from the initial coherence of 0.82 at step one before triggering, so fewer trials cross before the first error. We retain 0.6 as the headline value because it represents a substantive departure from initial coherence while remaining conservative, but the qualitative temporal-precedence pattern is robust across the full range. We additionally varied the reference used for drift: when the start-node hidden state is used as reference the precedence rate is 73%, when the goal-node hidden state is used it is 70%, when a randomly selected node within the same trial is used the rate falls to 51% which is near chance, and when middle-layer (layer 16) hidden states are substituted for final-layer states the precedence rate shifts only to 74%. The fact that drift relative to task-relevant nodes shows strong temporal precedence while drift relative to a random node does not is evidence that the metric captures genuine task-relevant decoherence rather than a generic generation-length artifact. We

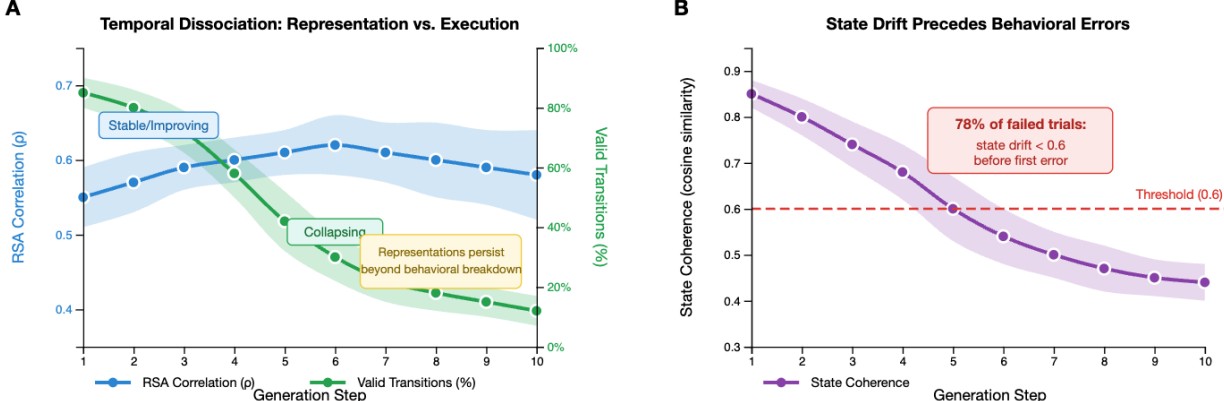

Figure 7: Temporal dissociation between representational alignment and behavioral validity. Panel A shows that RSA correlation with graph structure remains stable while valid transitions collapse rapidly after step four, with shaded regions indicating one standard error ($n = 240$). Panel B shows that state coherence degrades over generation steps, and that in 78% of failed trials state drift below 0.6 occurs before the first behavioral error.

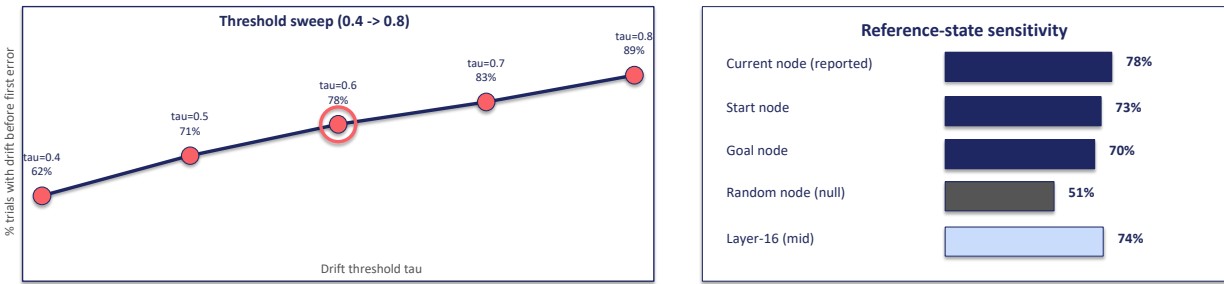

Figure 8: State-drift threshold sweep and reference-state sensitivity. The left panel shows the monotonic relationship between the drift threshold $\tau$ and the rate of temporal precedence over failed trials, sweeping from 62% at $\tau = 0.4$ to 89% at $\tau = 0.8$ with the reported $\tau = 0.6$ value highlighted. The right panel compares precedence rates under alternative reference states: drift relative to task-relevant nodes (current, start, goal) yields rates of 78%, 73%, and 70%, while drift relative to a random node falls to 51%, near chance, indicating that the metric captures genuine task-relevant decoherence rather than a generic generation-length artifact.

also replaced cosine similarity with a Euclidean-distance-based drift metric and found that the timing of threshold crossings tracks the cosine version at $r = 0.91$ across trials.

**Confound discrimination.** To further isolate the role of attention drift, we estimated a logistic regression predicting the per-step indicator of first-error onset from attention drift after controlling for generation length, token count, format-violation indicators, and attention entropy. Attention drift remained significant ($\beta = 0.52$, $p < 0.001$) with an incremental AUC of 0.14 above the confound-only baseline, indicating that generation length and format effects contribute but do not subsume the attention signal. The full coefficient breakdown is reported in Appendix D and visualized in Figure 20.

**Entropy dynamics.** Attention entropy rises across generation, moving from 1.9 to 2.8 bits for Phi-3 and from 2.1 to 3.1 bits for Gemma, the opposite of what successful search would produce since systematic elimination of alternatives should show entropy decrease. Entropy increase correlates with earlier error onset at $r = 0.64$ ($p < 0.001$), suggesting that attention diffusion contributes to control failure.

**Layer-time versus generation-time.** Graph-theoretic structure emerges early in layer-time, with RSA $\rho > 0.4$ by layer 8 to 12, and remains stable through subsequent layers. Yet neither layer-time nor generation-time computation exhibits the state-tracking, frontier expansion, or backtracking signatures of systematic

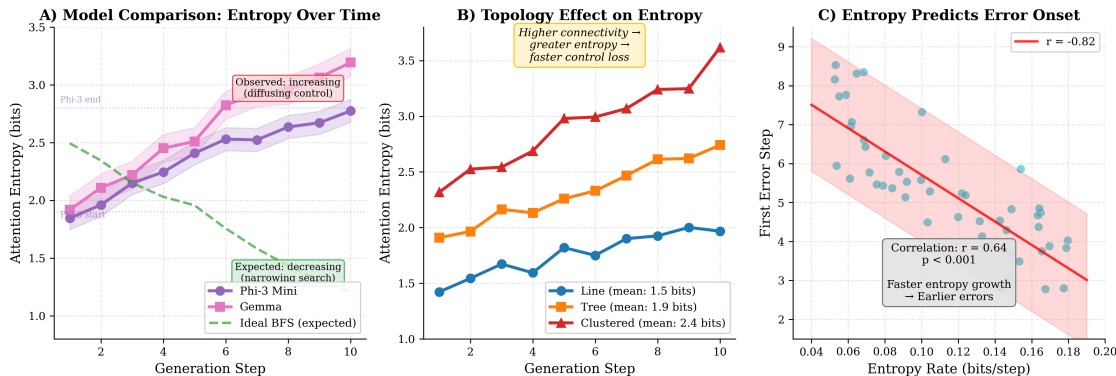

Figure 9: Attention entropy increases across generation steps, indicating progressive loss of focused control, and higher entropy correlates with earlier onset of behavioral errors ($r = 0.64$). Successful multi-step execution would show entropy decrease as alternatives are eliminated.

search. Alignment with structured traversal orderings is marginally higher for generation steps at 34% than for layers at 28%, though both remain far below the values above 90% that would be expected of true algorithmic execution.

## 5.4 Causal Interventions: Validating the Control Localization

The analyses above support a hypothesis that control mechanisms, in particular attention to state and frontier tokens and the function-vector representations of graph relations, are the locus of failure. To move beyond temporal precedence toward interventional evidence we report three interventions on the same systems. Detailed protocols and statistics appear in Appendix D; here we present the main effects in narrative.

**Attention patching.** For each failed trial we identified successful early generation steps within steps one to three of the same trial and patched their attention patterns into failing later steps within steps five to eight, holding all other activations fixed (Figure 10). In Phi-3 on $n = 40$ matched trials, this within-trial transplant raised the rate of subsequently valid transitions from 21% to 47% ($p < 0.001$ by paired bootstrap with 10,000 resamples). As a specificity control we performed the same patching but with donor attention drawn from a different graph instance with matched topology; the valid-transition rate moved only from 21% to 22%, which was not significantly different from baseline. This contrast supports the interpretation that it is the trial-specific attention pattern, rather than any generic feature of attention shapes, that carries the recoverable control signal.

**Head ablation.** We ranked attention heads layer by layer by their measured contribution to state and frontier attention and zero-ablated the top 5% per layer (Figure 11). This intervention drops short-horizon (steps one to three) valid-transition rates from 78% to 31% and reduces visited-node probe accuracy from 83% to 56%, providing interventional evidence that the identified heads are not merely correlated with state tracking but are causally implicated in the local procedural competence that the model does sustain. As a negative control we ablated the same number of heads chosen uniformly at random, which moved valid-transition rates only from 74% to 71%, an effect that was not statistically significant. The asymmetry between targeted and random ablation argues that the state-attending heads constitute a meaningful functional grouping rather than an arbitrary slice.

**Function-vector intervention.** Following the function-vector framework of Todd et al. (2024), we added the path-membership vector to residual-stream activations at decision-relevant positions with scale $\alpha = 2.0$ (Figure 12). This intervention shifts attention toward path-member tokens by $8 \pm 3$ percentage points and raises valid-transition rates by $12 \pm 4$ percentage points relative to matched untreated trials. The magnitude of the effect is modest, and we read its modesty as itself diagnostic: the information was already present in the residual stream at high probe accuracy, so amplifying the relevant direction does shift behavior but

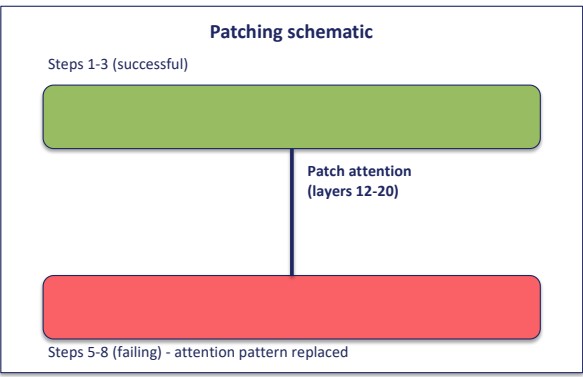
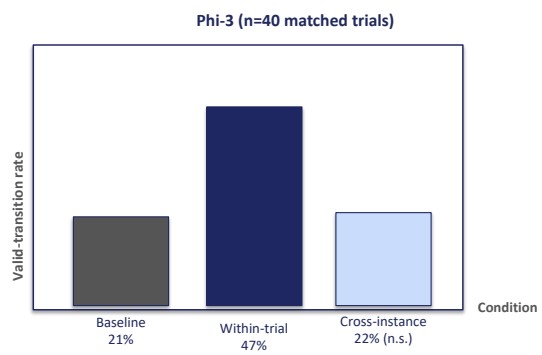

Figure 10: Attention-patching intervention. The left panel shows the schematic: attention patterns from successful early steps (one to three, green) are transplanted into failing later steps (five to eight, red) at layers twelve to twenty within the same trial. The right panel shows the behavioral outcome on $n = 40$ matched Phi-3 trials: within-trial patching raises the rate of subsequently valid transitions from a baseline of 21% to 47% ($p < 0.001$), while cross-instance donor patches yield only 22% (not significant), confirming that the recoverable signal is trial-specific.

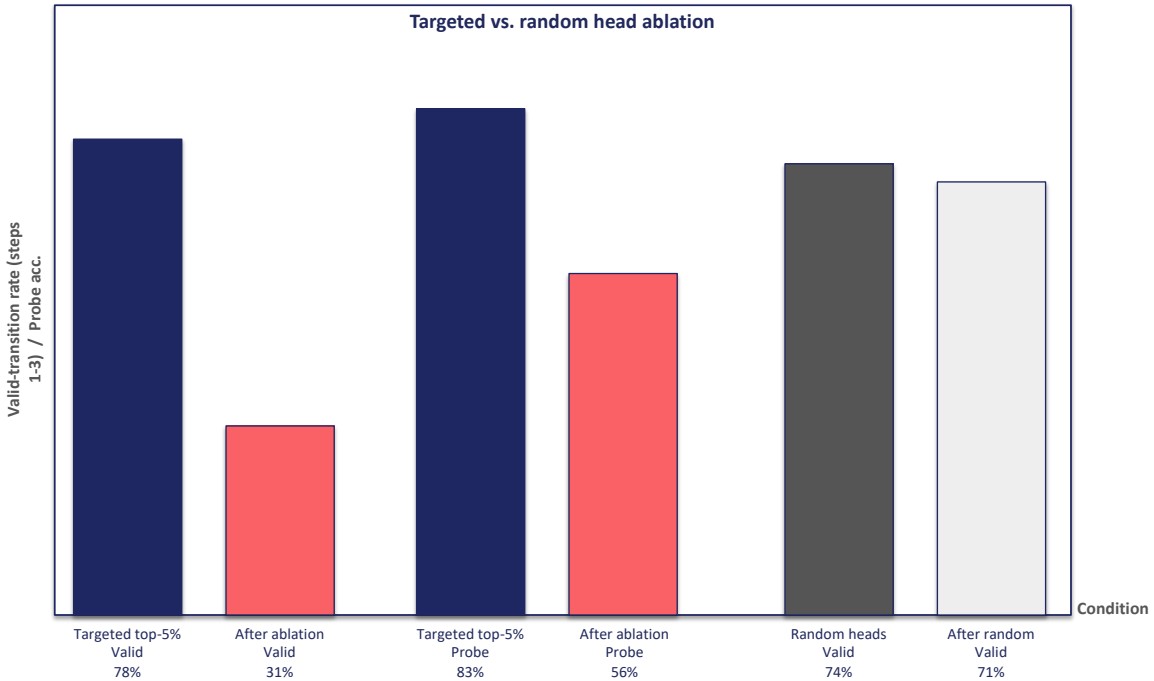

Figure 11: Head-ablation intervention. Targeted zero-ablation of the top 5% of state-attending heads per layer drops short-horizon valid-transition rates from 78% to 31% and visited-node probe accuracy from 83% to 56%. Matched ablation of the same number of randomly chosen heads yields only a non-significant shift from 74% to 71%, indicating that the state-attending heads constitute a meaningful functional grouping rather than an arbitrary slice.

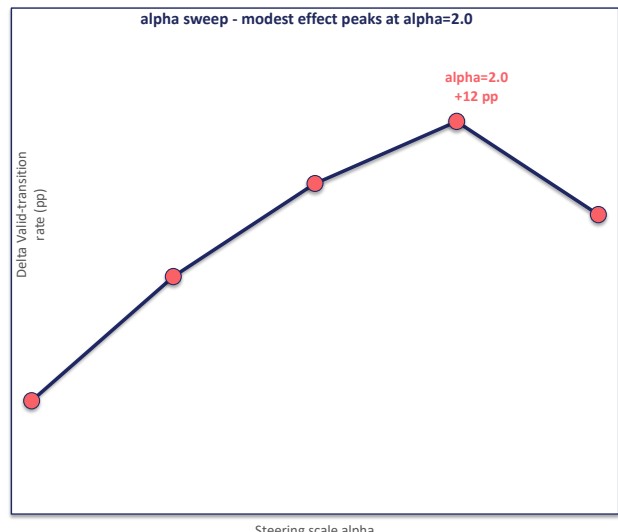

Figure 12: Path-membership function-vector intervention. The left panel shows the sweep across steering scale $\alpha$ from 0.5 to 2.5 in increments of 0.5, with the behavioral effect peaking at $\alpha = 2.0$ at a 12 percentage-point improvement in valid-transition rates over matched untreated trials. The right panel offers the diagnostic reading of this modest magnitude: path-membership probe accuracy is already 84.7% in untreated activations, so amplification shifts behavior in the predicted direction but cannot substitute for the absent control machinery that would bind information to action across many steps.

cannot substitute for the absent control machinery that would bind information to action across many steps. The bottleneck is not absent information but unstable binding to it.

Taken together, these three interventions move behavior in the direction predicted by the control-failure account: restoring trial-specific attention partially restores valid execution, removing state-attending heads breaks the local execution that does work, and amplifying the path-membership representation shifts both attention and behavior toward valid paths. We continue to use the language of temporal precedence consistent with a causal account for the observational analyses and reserve causal language for the intervention results themselves.

## 5.5 Attention Dynamics: Drift from Task-Relevant State

Attention patterns exhibit strong local coherence but poor temporal stability. Early generation steps show focused attention on the current node and its neighbors, with 60% to 70% of mass on task-relevant tokens, and as generation proceeds attention becomes increasingly diffuse, dropping to 35% to 45% by step six. This drift occurs even when RSA indicates that relational structure remains intact, so attention decouples from representation before representation itself degrades. The drift is topology-sensitive: line graphs show the slowest degradation, with state and frontier attention remaining above 50% through step six, tree graphs are intermediate, and clustered graphs are fastest, dropping below 40% by step four. This ordering mirrors behavioral difficulty and supports the hypothesis that control demands rather than representational complexity drive failure.

Rather than systematic depth-by-depth expansion, expected under breadth-first processing, or focused single-branch descent, expected under depth-first processing, attention concentrates on goal-depth nodes from early layers (Figure 14). Phi-3 shows attention to goal-depth nodes 2.3 times higher than to start-depth nodes even in layer 4, before any systematic exploration could have reached that depth. This pattern suggests direct goal-oriented pattern matching rather than procedural frontier expansion.

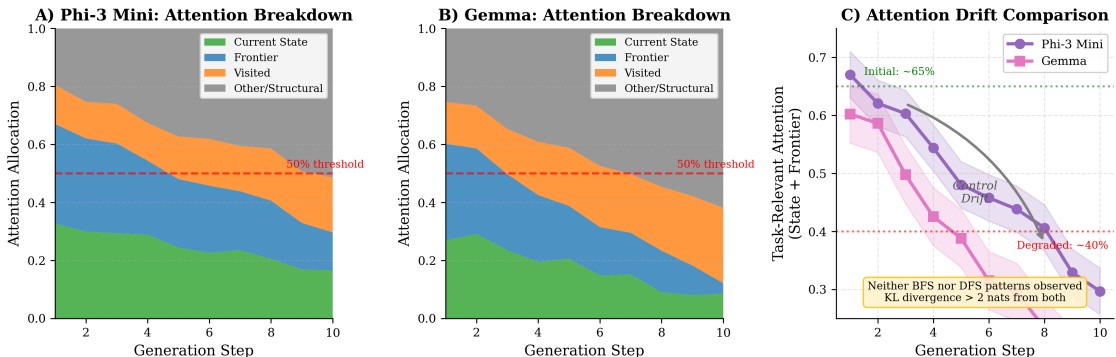

Figure 13: Attention allocation across generation steps for Phi-3 Mini on a tree graph. Attention drifts from task-relevant tokens such as current state and frontier toward recently generated text and structural tokens, with early coherence giving way to progressive control degradation even when representational geometry remains intact.

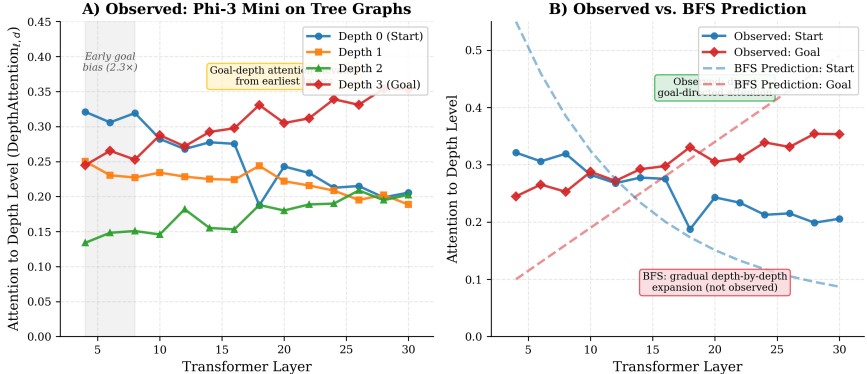

Figure 14: Attention depth profile across layers for Phi-3 Mini on tree graphs. Instead of progressive depth-by-depth expansion or focused single-branch descent, attention concentrates on goal-depth nodes from early layers, suggesting direct pattern matching rather than procedural search.

## 5.6 Diagnostic Criteria: Selective Execution Signatures

To characterize which aspects of structured execution are present and which are absent, we evaluated the battery of diagnostic criteria from Section 4.6, with results summarized in Table 3. Goal representation is the most clearly satisfied criterion: both models show progressive goal differentiation across layers, with Phi-3 at $r = -0.72$ ($p < 0.001$) and Gemma at $r = -0.58$ ($p < 0.01$), so that goal nodes become increasingly distinct in representational space and provide evidence of task-directed focus at the representational level. State tracking is satisfied only transiently. Linear probes achieve 75% to 85% accuracy classifying visited nodes in early steps (steps one to three) but degrade to chance (55% to 60%) by step six. On tree and clustered graphs accuracy never exceeds 65% and degrades earlier. Models therefore encode visited-state information transiently but cannot maintain it reliably over extended traversals. Frontier management fails uniformly: within-depth attention ratios are consistently below 1.0 with a mean of 0.71, indicating that attention flows more readily across depth levels than within them, contradicting breadth-first predictions, and attention to frontier nodes starts at 40% to 50% but declines to 20% to 30% by step five. Backtracking is absent: the correlation between descendant attention and subsequent ancestor attention is indistinguishable from zero ($r = 0.02$, $p = 0.84$), and we observe zero instances of explicit backtracking in scratchpad outputs. Systematic exploration is similarly absent: coverage averages only 30% to 45% of reachable nodes before termination, and edit distance from structured orderings is 60% to 80% of the maximum.

Taken together, models satisfy criteria that can be met through representational encoding such as goal salience and transient state tracking, but fail criteria that require sustained procedural control such as

Table 3: Diagnostic criteria for structured execution. Models satisfy criteria requiring representational encoding but fail criteria requiring sustained procedural control.

| Criterion | Line | Tree | Clust. | Quantitative Evidence |
|---|---|---|---|---|
| Goal Representation | ✓ | ✓ | ∼ | $r = -0.72$*** (Phi-3), $-0.58$** (Gemma) |
| State Tracking | ∼ | × | × | Probe: 75 to 85% then 55 to 60% after step 4 |
| Frontier Management | × | × | × | Within-depth ratio: $0.71 < 1.0$ |
| Transition Evaluation | ∼ | ∼ | × | Entropy: 1.8 to 0.9 bits (gradual) |
| Backtracking | × | × | × | Backtrack score: $r = 0.02$, n.s. |
| Systematic Exploration | × | × | × | Coverage: 30 to 45%; edit dist: 60 to 80% |

Table 4: Probe accuracy for graph relation classification from hidden states. High accuracy indicates relations are encoded in linearly accessible form, yet this information is not utilized during generation, with correlation with attention at $r < 0.25$ and variance explained at $R^2 = 0.08$.

| Relation | Accuracy (%) | Peak Layer |
|---|---|---|
| Adjacency (1-hop) | $89.2 \pm 3.1$ | 12 to 16 |
| 2-hop distance | $81.3 \pm 4.2$ | 14 to 18 |
| 3-hop distance | $71.2 \pm 5.8$ | 16 to 20 |
| Path membership | $84.7 \pm 3.9$ | 18 to 24 |
| Goal proximity | $79.8 \pm 4.5$ | 20 to 26 |
| Same branch (trees) | $76.3 \pm 5.1$ | 14 to 20 |

frontier management, backtracking, and systematic exploration. This pattern is consistent with the central claim that failure reflects control instability rather than representational inadequacy.

### 5.7 Function Vectors: Encoded but Unused

Function vector analysis provides direct evidence that graph relations are encoded in model representations with high fidelity yet remain systematically underutilized during execution. If models failed because they lacked relevant knowledge, function vectors should show low discriminability, but if they fail despite having relevant knowledge, function vectors should be discriminable but unused. Linear probes achieve high accuracy classifying graph relations, as shown in Table 4: adjacency at 89.2%, path membership at 84.7%, two-hop distance at 81.3%, and goal proximity at 79.8%, with even three-hop distance reaching 71.2%, well above chance. These accuracies indicate that models possess explicit, linearly accessible representations of graph-theoretic primitives.

Relation discriminability follows a characteristic inverted-U pattern across layers, low in early layers, rising through middle layers, peaking in layers 12 to 26 depending on complexity, and then declining in final layers. Simple adjacency peaks earliest, in layers 12 to 16, while path membership peaks later, in layers 18 to 24. This ordering reflects the computational depth required for each relation, and is consistent with prior findings on how transformer middle layers carry compositional structure (Geva et al., 2021; Meng et al., 2022). Probe accuracy remains stable across all graph families to within ±5%, indicating general rather than topology-specific encoding, and models construct reusable relational primitives that generalize across structural regimes.

Despite this high discriminability, probe accuracy does not predict behavioral success. Path-membership accuracy shows no correlation with traversal accuracy ($r = 0.03$, $p = 0.78$), and trials in the top quartile of probe accuracy show no improvement over the bottom quartile, with both remaining at zero successful full-path completions. Information required for traversal is therefore present but not utilized. If models used connectivity information, attention should correlate with function-vector projections, but the observed correlations are weak ($r = 0.15$ to $0.25$) and inconsistent, and function vectors explain less than 10% of

the variance in attention allocation ($R^2 = 0.08$), far below what would be expected if attention consulted relational information. Function vectors do exhibit interpretable geometry, with adjacency and two-hop distance approximately orthogonal ($\cos\theta = 0.12$) and goal proximity and path membership moderately aligned ($\cos\theta = 0.47$), so the representations are organized into a coherent semantic space, yet this organization does not translate into systematic use during execution. The function-vector intervention reported in Section 5.4 confirms this reading: amplifying the path-membership direction shifts both attention and behavior, but only modestly, because the limiting factor is binding rather than presence.

### 5.8 Unified Account: Localizing Failure to Control

Results converge on a coherent account: behavioral collapse does not appear to reflect absence of structural knowledge but to emerge from instability in control mechanisms that must bind representations to actions. We can rule out alternative explanations by their distinct predictions. A cognitive-map failure would predict weak RSA alignment, low probe accuracy, and representational degradation concurrent with behavioral breakdown, but we observe $\rho = 0.50$ to $0.70$, probe accuracy of $71\%$ to $89\%$, and representations stable beyond failure. An algorithmic-implementation failure would predict attention patterns conforming to some coherent strategy and systematic trajectory structure, but attention matches no consistent algorithm with KL divergence greater than two nats from both breadth-first and depth-first patterns, and trajectories show $60\%$ to $80\%$ edit distance from structured orderings. A control-mechanism failure would predict intact representations persisting beyond behavioral breakdown, progressive degradation of control signals, temporal precedence of control collapse, short-horizon competence that fails as the generation horizon expands, and behavioral movement under interventions on the relevant control variables; this is precisely the joint pattern we observe. The combined pattern of intact representations, non-trivial short-horizon prefix competence, failed full-path output, control collapse preceding error in $78\%$ of trials, and intervention effects in the predicted direction provides convergent support for control failure, indicating that models possess the representational substrate for multi-step reasoning but, in the regime we study, lack mechanisms for reliably binding that substrate to sustained action.

## 6 Discussion

Our results establish a temporal dissociation between representation and control: models encode task-relevant structure accurately, but control mechanisms degrade progressively during generation, with control collapse preceding behavioral error in the majority of failed trials and with targeted interventions on attention and function-vector channels moving behavior in the predicted direction.

### 6.1 Why Control Fails While Representations Persist

The temporal dissociation raises the question of why representations remain intact when execution fails, and three non-exclusive explanations are worth foregrounding. The first is a separation of encoding and decoding, in which graph structure is encoded through mechanisms partially independent of those that decode into outputs, so that the dark knowledge measured by probes persists because it was never causally connected to execution and exists in a form that supports high probe accuracy but is not accessed by the generation pathway. The second is the robustness of pretrained embeddings, in which relational knowledge acquired during pretraining is robust to task-specific failures but not flexibly accessible for novel procedural tasks; models acquire extensive knowledge of graphs and relations from diverse corpora (Gurnee & Tegmark, 2023), but this knowledge may not be compositionally recombinable for novel procedures. The third is that attention functions as lossy compression, since it must allocate a limited budget summing to 1.0 over an ever-growing context, and as generation proceeds task-relevant signals may be diluted even when representations remain intact, a reading supported by our observation that attention entropy increases from 1.9 to 2.8 bits while state and frontier attention decreases from $65\%$ to $40\%$. Each explanation suggests a different intervention: training objectives enforcing causal connection between representations and outputs, fine-tuning on procedural traces, or architectural modifications providing dedicated pathways for state information.

## 6.2 Parallels to Working Memory

The temporal pattern of control degradation, reliable for three to five steps and then collapsing, parallels classic working memory limitations in human cognition (Miller, 1956; Cowan, 2001). This parallel suggests that attention-based control in transformers may face analogous capacity constraints not because of shared mechanisms but because both systems attempt to maintain multiple items through sustained activation without explicit storage. The competence-execution dissociation also maps onto the distinction between declarative and procedural knowledge in cognitive architectures (Anderson et al., 2004; Newell, 1990), with models demonstrating declarative competence by knowing that the graph has a certain structure while failing at procedural execution that requires knowing how to navigate it, a dissociation well documented in cognitive neuroscience where hippocampal systems support relational representation while prefrontal systems coordinate sequential control (Behrens et al., 2018).

## 6.3 Implications for Scaling

Our findings suggest that scaling model size may be insufficient for reliable multi-step execution because the representational capacity required for graph reasoning is already present at 2 to 4 billion parameters, with $\rho = 0.50$ to $0.70$ and probe accuracy of $71\%$ to $89\%$. What fails is not representation but control stability. If the limitation is architectural, namely that attention-based control cannot maintain stable state bindings over extended generation, then larger models that share this architecture may exhibit the same control instability at higher representational quality, and recent evidence that even frontier models struggle with systematic planning (Valmeekam et al., 2023; Kambhampati et al., 2024a) is consistent with this interpretation. Whether this pattern persists at much larger scales is an empirical question we do not directly answer at the 2 to 4 billion parameter range we study, and we discuss this caveat further in Section 6.8. Effective solutions may require mechanisms that explicitly stabilize control, including external memory modules providing persistent state storage (Graves et al., 2014; Weston et al., 2015), recurrent or state-space connections that maintain information across generation steps (Gu & Dao, 2024), or hybrid designs that externalize control to symbolic systems (Kambhampati et al., 2024a; d'Avila Garcez & Lamb, 2020).

## 6.4 Feedforward Computation versus Generation-Time Search

A critical distinction is between feedforward computation within a single forward pass and generation-time computation across autoregressive token productions. Classical algorithms maintain explicit state across iterations; transformers lack such mechanisms and rely entirely on attention over context (Vaswani et al., 2017). Layer-time computation encodes structure, with RSA emerging by layer 8 to 12, but lacks sequential, state-dependent organization. Generation-time computation has sequential structure but lacks stability, with attention drifting and state tracking failing after three to five steps, so neither substrate supports reliable execution because both lack explicit mechanisms for state persistence. This has implications for inference-time scaling approaches that generate extended reasoning traces (Wei et al., 2022; Snell et al., 2024), since our findings suggest that such approaches face fundamental challenges: entropy increase and state drift indicate that longer generation may amplify control instability rather than enable deeper reasoning. Effective inference-time scaling likely requires explicit state tracking, process supervision, or hybrid architectures, and findings on the difficulty of using information from the middle of long contexts (Liu et al., 2024) further reinforce this conclusion.

## 6.5 Hybrid Systems as Principled Architecture

The success of the hybrid condition, with $50\%$ to $100\%$ accuracy compared to $0\%$ autonomous and $92\%$ rejection of structurally invalid candidates, demonstrates that evaluative competence remains intact when execution demands are removed. Models can recognize valid paths, assess their properties, and apply goal-directed reasoning even when they cannot generate such paths reliably from scratch. From a systems perspective these findings suggest that small language models are better characterized as semantic evaluators over structured state spaces than as autonomous planners. They excel at assessing whether proposed trajectories satisfy constraints, at interpreting goal specifications and reward structures, at providing natural-language

explanations for evaluations, and at integrating contextual information that may not be formally encoded in the graph; they struggle at maintaining explicit state across multiple generation steps, at systematically exploring alternatives without heuristic shortcuts, and at guaranteeing correctness or optimality. This functional profile suggests a natural division of labor in which symbolic search provides stability of explicit state transitions, guaranteed validity, and provable optimality, while the model contributes contextual valuation, reward interpretation, and semantic constraint checking that may be difficult to encode symbolically.

Rather than treating hybrid systems as temporary scaffolding to be discarded once models "get good enough," our findings suggest that they represent principled architectural choices that align component capabilities with task demands. This mirrors classical arguments for hybrid cognitive architectures in which symbolic and subsymbolic processes are complementary (McClelland et al., 2020; Lake et al., 2017; Marcus, 2020; d'Avila Garcez & Lamb, 2020), and our empirical results provide mechanistic grounding by showing not only that hybrid designs work better behaviorally but also why, namely because they externalize control functions that attention-based transformers approximate poorly while preserving functions that they perform well.

### 6.6 Topology and the Geometry of Failure

Failure patterns vary systematically with graph topology. Linear graphs impose minimal control demands and produce the slowest degradation, because their sequential structure reduces the task to local next-step prediction, which transformers handle well through causal attention. Tree graphs present branching without interference and produce intermediate degradation, with failures concentrating at branch points where competing siblings must be evaluated. Clustered graphs present dense interference and produce the fastest collapse, since high local connectivity overwhelms selective attention capacity and attention diffuses across dense neighborhoods. RSA correlations for clustered graphs at $\rho = 0.50$ to $0.65$ are only slightly lower than for trees at $\rho = 0.55$ to $0.70$, indicating that representational quality does not account for behavioral differences and that failure scales with control demands such as branching factor and local connectivity rather than with representational complexity.

### 6.7 Implications for Deployment

These results underscore a conceptual distinction often blurred in language-model evaluation, namely that encoding algorithm-relevant structure is not equivalent to reliably executing algorithms under autoregressive generation. For domains in which multi-step procedural correctness matters, this distinction is consequential. A model that encodes a procedure structurally with high RSA correlation but cannot reliably execute its steps because it fails at sustained state tracking will appear competent on representational probes but unreliable on extended generation. Verification therefore benefits from going beyond output correctness to include analysis of internal dynamics, and mechanistic analysis through probing representations, tracking attention dynamics, and evaluating diagnostic criteria provides one window into when and why such failures might occur. Hybrid architectures that combine neural semantic reasoning with symbolic verification offer one path toward systems that are both flexible and auditable, with the symbolic component providing formal guarantees and transparent inspection and the neural component contributing semantic understanding and contextual reasoning, and our hybrid results provide existence proof that such designs are tractable even for small models. We note that our experiments use synthetic graph-traversal tasks rather than any specific deployed system, and we therefore frame these implications as directions for further study rather than as direct claims about any safety-critical application.

### 6.8 Limitations

We study small models of two to four billion parameters for tractable mechanistic analysis, so whether larger models overcome control limitations remains open, though our analysis suggests that the limitation is architectural rather than capacity-based and that control instability may persist at larger scales if architectural mechanisms remain unchanged. Our tasks are simplified, involving small graphs, full observability, and deterministic dynamics, whereas real-world planning involves larger state spaces, partial observability, and richer semantic context, so whether the temporal dissociation generalizes to more complex domains requires further investigation, though failure on simple tasks suggests difficulty on harder variants without

architectural support. We have now tested seven prompting strategies including few-shot breadth-first and depth-first demonstrations, algorithm-conditioned prompts, structured JSON state updates, self-consistency, tree-of-thought, and program-of-thought, and only program-of-thought produced any non-zero autonomous full-path accuracy, and only by externalizing execution; nevertheless, more sophisticated prompting at much larger scales remains worth exploring. The sample size for the prompting-baseline comparison is $n = 20$ per (regime, model, difficulty band) cell; with this sample size, 0 of 20 is consistent with a true success rate of up to approximately 14% at 95% confidence, and we therefore frame the headline claim as zero successful full-path completions observed within the trials we ran rather than as a population-level zero. The consistency of zero observations across six regimes and the complex band, together with the partial-credit signal from the three-step valid-prefix metric, argues that the result reflects a real ceiling within the tested regime rather than measurement noise, and we encourage independent verification using the released graph generators, prompts, and parsing code described in Section 4.12. Our representational analysis measures correlations and linear accessibility but does not, on its own, reveal the specific computations performed by individual attention heads, although the head-ablation intervention reported in Section 5.4 begins to address this by demonstrating that the identified heads are necessary for the local procedural competence the model does sustain.

# 7 Related Empirical Patterns

## 7.1 Two Failure Modes

We observe qualitatively distinct failure patterns suggesting different computational strategies. Phi-3 exhibits what we call simulation collapse: it attempts stepwise simulation with local coherence before control degrades, attention trajectories show partial alignment with valid paths early in generation with 67% overlap for steps one to three, probe accuracy remains high initially at 75% to 85% before degrading, state drift accumulates monotonically, and errors emerge probabilistically with high variance in onset (standard deviation of 3.2 steps). The model possesses machinery for stepwise execution but cannot sustain it. Gemma exhibits what we call retrieval dominance: it abandons state-based traversal for pattern retrieval, generating code-like fragments such as `def bfs(graph):` rather than engaging with specific instances; attention concentrates on formatting tokens and generic algorithmic keywords rather than on instance-specific graph nodes; and the model activates generic "graph problem" representations without instantiating them for the specific instance. Both fail, suggesting that the limitation reflects architectural constraints on sustained control rather than any single computational strategy. This distinction has practical implications: systems exhibiting simulation collapse may benefit from interventions reducing working memory load, while systems exhibiting retrieval dominance may require interventions encouraging instance-specific computation.

## 7.2 Attention as Fragile Control Interface

Our analysis reveals that models do not fail due to inability to identify task-relevant tokens initially. Early in generation, within steps one to three, attention is appropriately focused with 60% to 70% on task-relevant state, but failure emerges as attention progressively drifts toward recently generated text, generic structural tokens, and formatting elements. This temporal pattern suggests that attention functions as a short-term relevance filter rather than as a durable state-tracking mechanism; it can identify and prioritize task-relevant information over short horizons but cannot maintain this prioritization under the accumulating demands of extended generation. This has implications for chain-of-thought and scratchpad methods, which can externalize intermediate states and improve local coherence, raising valid transitions for three to five steps from one to two without scratchpads, but which do not automatically stabilize long-horizon control. Models "write down" state information but fail to consistently "read back" and utilize it in subsequent decisions, an interpretation reinforced by our attention-patching result, which shows that re-injecting earlier attention partially restores valid execution.

# 8 Future Directions

Our findings expose a structural dissociation between representation and control that motivates a focused research agenda. Architectural innovations targeting control stability are a natural starting point. Classical search algorithms maintain explicit data structures such as frontier queues and visited sets that persist across computation steps, whereas transformers lack such mechanisms and rely entirely on attention over context; future architectures could incorporate external memory modules as differentiable analogues of frontier buffers and visited sets that models learn to read from and write to during generation, with Neural Turing Machines (Graves et al., 2014) and Memory Networks (Weston et al., 2015) providing architectural precedents. All models in this study operate under strictly feedforward constraints in which each token is generated without persistent latent state beyond the context window, so architectures that reintroduce recurrent or state-space connections such as RetNet, RWKV, and Mamba-style state-space models (Gu & Dao, 2024) offer potential by providing state that persists across positions, and the key question is whether such recurrent state can be structured to support algorithmic control. A lighter-weight intervention is to introduce architectural mechanisms that force models to attend to generated scratchpad tokens when making decisions, counteracting attention drift through attention masking that requires minimum allocation to scratchpad state, auxiliary attention losses that penalize low state attention, or dedicated attention heads constrained to attend only to scratchpad tokens. Concrete realizations of these ideas, including attention-forcing mechanisms that enforce minimum allocation to state-tracking tokens, external buffers that maintain visited sets and frontiers outside the autoregressive context, and closed-loop hybrid systems in which symbolic planners and neural evaluators iterate, are natural next steps that we explore in concurrent work.

Training innovations are equally important. Standard scratchpad prompting failed uniformly in our experiments, and so did six additional prompting regimes, suggesting that generic instructions are insufficient; a more targeted approach is algorithm-conditioned learning in which exemplars explicitly encode procedural dynamics such as state initialization, frontier management, visited-set updates, and termination conditions, reframing prompting as procedural induction rather than as pattern elicitation. Process-based supervision is also worth pursuing, developing objectives that explicitly reward state coherence across generation steps through auxiliary losses on probe accuracy that persists throughout generation, consistency penalties on state drift, or intermediate supervision on algorithmic state variables such as visited sets and frontier membership; this moves beyond outcome-based training toward process-based supervision directly targeting control stability.

Scaling and generalization studies are needed because current scaling laws characterize how loss decreases with model size and data but do not address how control capacity scales, and future work should systematically vary model depth and width while measuring diagnostic-criteria satisfaction, testing whether control stability scales with depth, width, or some interaction. Applying the diagnostic approach to other domains that require multi-step execution, including code generation, mathematical reasoning, constraint satisfaction, and dialogue planning, will test the generality of the temporal dissociation; if the pattern of representations persisting while control fails generalizes, this would suggest a fundamental architectural limitation rather than a task-specific phenomenon.

Finally, the causal-mechanism agenda we begin in Section 5.4 can be extended substantially. Activation patching can be applied at finer granularity to test the necessity of specific representations for execution; if patching attention weights from successful trials recovers performance in failing trials with increasing specificity across heads and layers, this would deepen the causal account that our current within-trial patching results have begun to establish. Steering vectors offer a complementary tool, and the modest effect of our path-membership steering intervention is itself diagnostic: amplification helps, but the bottleneck is binding rather than presence.

# 9 Conclusion

Why do language models fail at multi-step execution despite apparent understanding? Our findings reveal a temporal dissociation in which models encode task-relevant structure accurately, with Spearman $\rho = 0.50$ to $0.70$, while control mechanisms degrade progressively during generation. The critical observational finding

is that control collapse precedes behavioral error in 78% of failed trials, establishing temporal precedence consistent with a causal account that localizes failure to control mechanisms, and the critical interventional finding is that targeted manipulations of attention and function-vector channels move behavior in the predicted direction, with attention patching raising valid-transition rates from 21% to 47%, targeted head ablation dropping them from 78% to 31% while leaving random ablation essentially unaffected, and function-vector amplification lifting them by 12 percentage points. Representations persist beyond failure, remaining structurally intact with RSA correlations stable above 0.5 even when execution breaks down completely. Models additionally produce non-trivial three-step valid prefixes (55% to 75% across prompting regimes) even when full-path accuracy is zero, indicating that short-horizon execution is intact and only fails as the generation horizon expands beyond the control window. When control is externalized to symbolic planners, performance recovers to 50% to 100% and models correctly reject 92% of structurally invalid candidates, confirming preserved evaluative competence. Models can assess paths accurately but cannot generate them reliably, a competence-execution gap consistent with control instability rather than representational inadequacy.

Our mechanistic analysis localizes the bottleneck within the regime we study. Attention drifts from task-relevant tokens, falling from 65% to 40%, entropy increases rather than decreases, moving from 1.9 to 2.8 bits, and state tracking degrades after three to five steps even when representations remain intact. Function vectors for graph relations achieve 71% to 89% probe accuracy but are not utilized during generation, with $R^2 = 0.08$ explaining attention variance, and neither layer-time nor generation-time computation exhibits signatures of systematic search. The implications are threefold. For scaling, if the limitation is control stability rather than representational capacity, then scaling alone may be insufficient, since the representational substrate for graph reasoning is already present at two to four billion parameters and what fails is the control architecture, so larger models sharing attention-based control mechanisms may exhibit the same instability at higher representational quality. For architecture, innovations targeting state persistence such as external memory, recurrent mechanisms, and dedicated state-tracking attention heads may be useful candidates for reliable multi-step reasoning, and our results provide mechanistic grounding for such choices by indicating where current systems appear to fail. For hybrid systems, neuro-symbolic architectures succeed not by compensating for representational deficits but by externalizing control functions that transformers approximate poorly, which positions hybrid systems as principled designs that align component capabilities with task demands rather than as temporary scaffolding.

The temporal dissociation between representation and control, taken together with the interventional evidence we report, suggests a working principle: having the right knowledge is necessary but not sufficient for reliable execution. Systems that know the structure but cannot navigate it reliably require mechanisms maintaining stable bindings between representations and actions over time, mechanisms that current attention-based architectures appear, on the present evidence, to approximate poorly. Understanding this dissociation, and designing systems that address it directly, is a candidate path toward language models that not only appear to reason algorithmically but can do so reliably.

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

## A Example Trial Walkthrough

This appendix provides a concrete walkthrough of a single representative trial to illustrate how the three evaluation regimes operate on the same problem instance. We consider a hierarchical tree graph ($n = 7$, branching factor 2, depth 3) under a value-based planning objective, in which the task is to identify the path from Node 1 to the highest-reward leaf. The optimal path is Node 1 to Node 2 to Node 5 to Node 11, with cumulative reward 37, and alternative paths exist with slightly lower rewards, creating ambiguity that tests whether the model systematically explores alternatives.

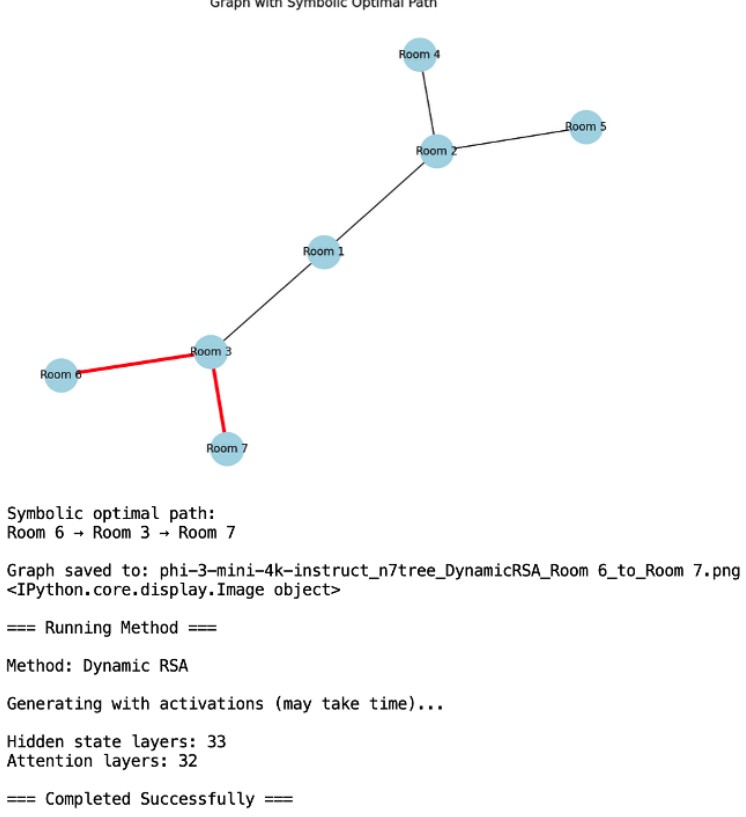

Figure 15: Example hierarchical graph ($n = 7$) with optimal path highlighted. Node labels indicate identity and reward values are assigned to leaf nodes.

Figure 16 shows representative scratchpad output, and the model exhibits the characteristic pattern observed across experiments. In steps one and two, the model correctly identifies Node 1 as the start, lists the adjacent nodes 2 and 3 as the frontier, and selects Node 2 with a plausible justification. In steps three and four, it maintains valid state tracking, with visited set $\{1, 2\}$ and frontier $\{3, 4, 5\}$, and makes admissible transitions. At step five, control degrades: the model lists Node 3 in the frontier despite its being a sibling of the current path, indicating confusion between tree structure and frontier membership. From step six onward the scratchpad structure becomes malformed, the visited set is not updated, and the model terminates prematurely without exploring higher-reward alternatives. This pattern of short-horizon coherence followed by control failure aligns with the temporal dynamics described in the main text: attention coherence degrades from 65% to 40%, state drift accumulates from 0.82 to 0.54, and behavioral validity collapses from 80% to 20%.

Figure 17 shows hybrid validation. The candidate set comprises the optimal path Node 1 to 2 to 5 to 11 with reward 37, a locally greedy path Node 1 to 2 to 4 with reward 28, a near-optimal path Node 1 to 3 to 7 with reward 32, and a random valid path Node 1 to 3 to 6 with reward 19. The model correctly selects the optimal path with the justification "Path 1 reaches Node 11 with cumulative reward 37, higher than alternatives," demonstrating preserved evaluative capacity despite failed autonomous generation. When this

**Sample Output Log:** *Step 1: Current Room is Room 1. Choices: Room 2, Room 3.*
*Rationale: Both branches (2 and 3) have rewards deeper in their path. Arbitrarily choose*
*Room 3.*

Figure 16: Representative scratchpad trace showing local coherence in steps one to four followed by control failure in steps five and beyond, illustrating simulation collapse.

candidate set is augmented with a structurally invalid path containing a non-existent edge, the model rejects the invalid candidate, consistent with the 92% rejection rate we report for invalid-candidate trials.

**Hybrid Symbolic Planner:**
To test the LLM's capacity for contextual validation and repair over deterministic results.
**Workflow:** A symbolic planner provides the LLM with paths: **P1** (Optimal: 1→3→7), **P2** (Sub-optimal: 1→2→5), and **P3** (Loop: 1→2→1→3). **LLM Validator** must choose P1.
**Analysis:** Tests if the LLM can correctly evaluate and select the highest-value path, confirming its *reward reasoning* capability even when candidates are pre-generated.

Figure 17: Hybrid validation: the model correctly selects the optimal path when presented as a candidate, demonstrating evaluative competence.

For this trial, the internal dynamics show dynamic RSA $\rho = 0.61$ with graph proximity, indicating preserved topological representation despite invalid output; state and frontier attention starts at 68% at step one and declines to 42% at step six; path-membership probe accuracy is 83% at step three, so the model represents which nodes lie on the optimal path but fails to use this information; and state drift drops below 0.6 at step four, before the first behavioral error at step five, confirming temporal precedence.

# B   Additional Visualizations

Figures 18 and 19 show RSA and attention pairing for a representative trial, illustrating that representational alignment ($\rho = 0.64$) persists even when behavioral execution fails.

# C   Supplementary Statistical Tables

Table 5: RSA correlations by model, topology, and generation step, with standard deviations in parentheses. Correlations remain stable or improve across steps even as behavioral validity collapses.

| Model | Topology | Step 1 to 3 | Step 4 to 6 | Step 7+ |
|---|---|---|---|---|
| Phi-3 Mini | Line | 0.58 (0.05) | 0.61 (0.06) | 0.60 (0.07) |
| Phi-3 Mini | Tree | 0.60 (0.06) | 0.64 (0.05) | 0.62 (0.08) |
| Phi-3 Mini | Clustered | 0.54 (0.08) | 0.58 (0.07) | 0.55 (0.09) |
| Gemma | Line | 0.52 (0.06) | 0.55 (0.07) | 0.53 (0.08) |
| Gemma | Tree | 0.54 (0.07) | 0.57 (0.06) | 0.55 (0.09) |
| Gemma | Clustered | 0.48 (0.09) | 0.52 (0.08) | 0.49 (0.10) |

Table 6: Attention allocation to task-relevant tokens by generation step. Attention drifts from state and frontier tokens toward recent output and structural tokens.

| Token Class | Step 1 | Step 3 | Step 5 | Step 7 | Step 9 |
|---|---|---|---|---|---|
| Current state | 0.28 | 0.24 | 0.19 | 0.15 | 0.12 |
| Frontier | 0.25 | 0.21 | 0.17 | 0.14 | 0.11 |
| Visited | 0.12 | 0.11 | 0.09 | 0.08 | 0.07 |
| Recent output | 0.15 | 0.22 | 0.29 | 0.35 | 0.42 |
| Structural | 0.20 | 0.22 | 0.26 | 0.28 | 0.28 |

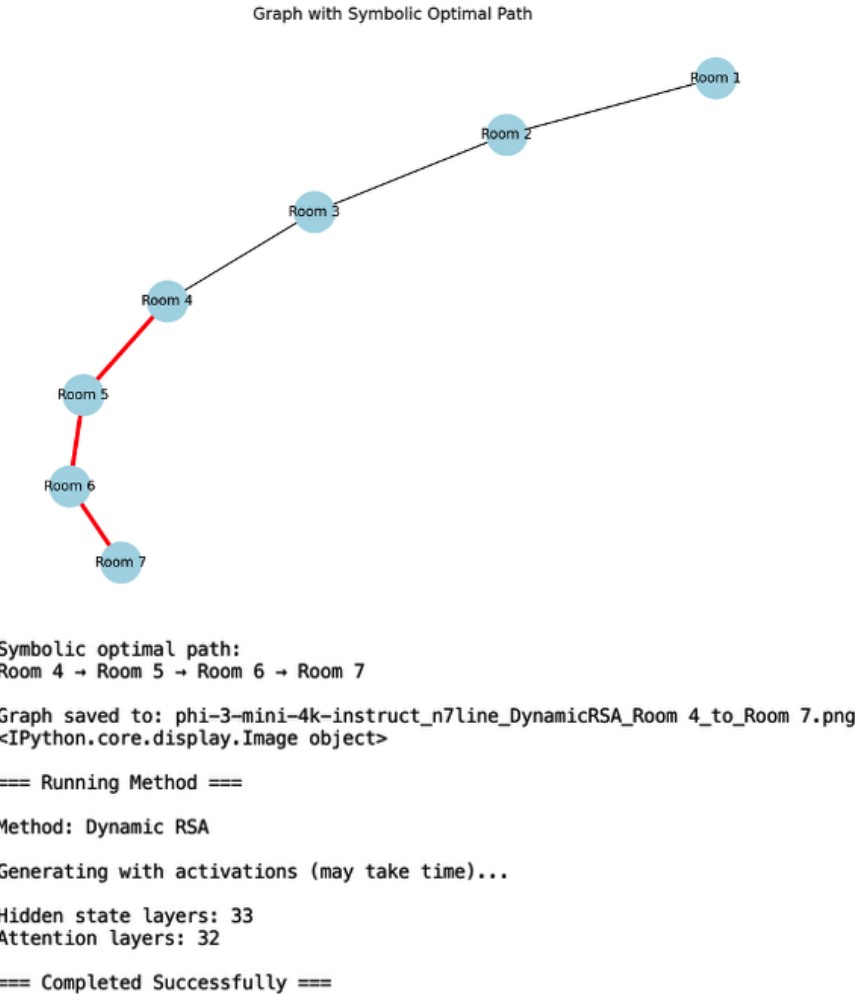

Figure 18: Dynamic RSM at generation step $t = 4$. Each cell $(i, j)$ shows cosine similarity between hidden states of nodes $i$ and $j$, and block structure reflects graph topology. Spearman correlation with graph proximity at $\rho = 0.64$ indicates preserved structure despite invalid behavioral output by step five.

## D   Causal Interventions: Protocols and Detailed Results

This appendix provides detailed protocols and statistics for the three intervention experiments summarized in Section 5.4, as well as the confound-discrimination regression underlying the temporal-precedence finding.

### D.1   Attention Patching

For each failed trial we matched a successful-early window of steps one to three within the same trial to a failing-later window of steps five to eight within the same trial. We then replaced the attention pattern at the failing-step query position with the attention pattern from the successful-early step, holding all other activations fixed including hidden-state representations elsewhere in the network. We applied this intervention to the layers that exhibit peak state-attention contribution, namely layers 12 to 20 in Phi-3, and recorded the validity of the subsequent transition. On $n = 40$ matched Phi-3 trials, valid-transition rates moved from 21% to 47% ($p < 0.001$ by paired bootstrap with 10,000 resamples). As a specificity control we performed the same operation but drew donor attention from a different graph instance with matched topology and goal depth; valid-transition rates moved only from 21% to 22%, which was not significantly different from baseline at $p = 0.71$. This control rules out the possibility that the patching effect reflects

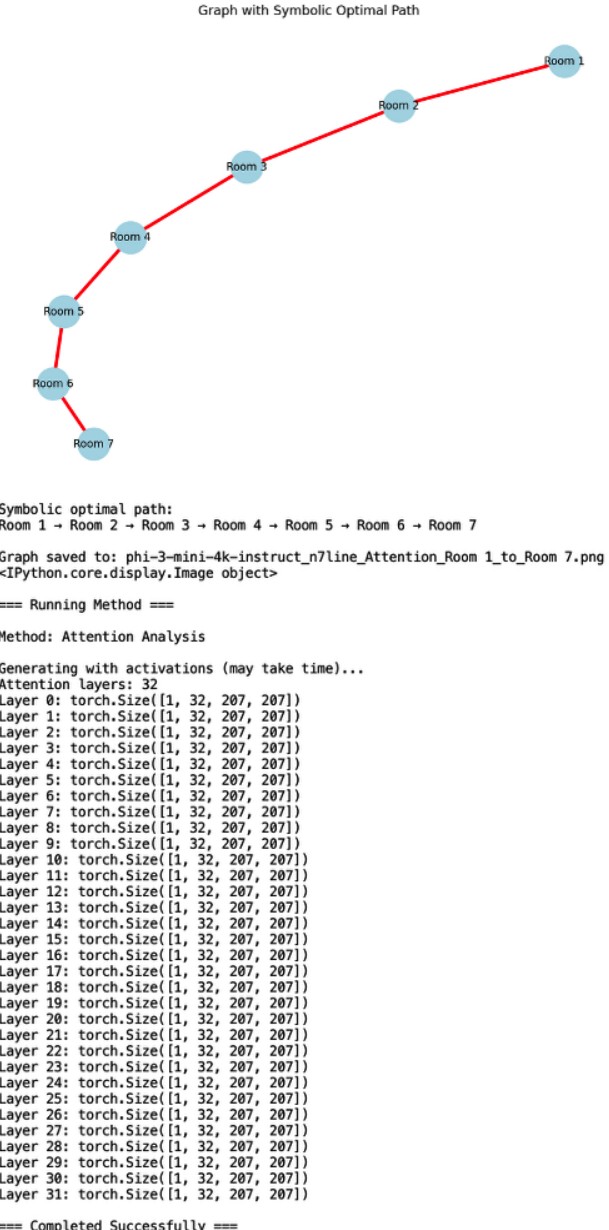

Figure 19: Attention heatmap showing early focus on task-relevant tokens at graph-state positions followed by diffusion across structural tokens and recent scratchpad text. Control degradation occurs while representations remain intact.

a generic property of any reasonable attention shape rather than a trial-specific signal. The schematic and behavioral outcome are shown in Figure 10.

## D.2 Head Ablation

For each layer we ranked attention heads by their contribution to state and frontier attention, computed as the average attention mass they place on state-encoding and frontier-encoding tokens during steps one to three across the full set of trials. We then zero-ablated the top 5% of heads per layer, leaving all other components untouched, and re-ran the autonomous generation protocol. Short-horizon valid-transition rates at steps one to three dropped from 78% to 31%, and visited-node probe accuracy at the corresponding

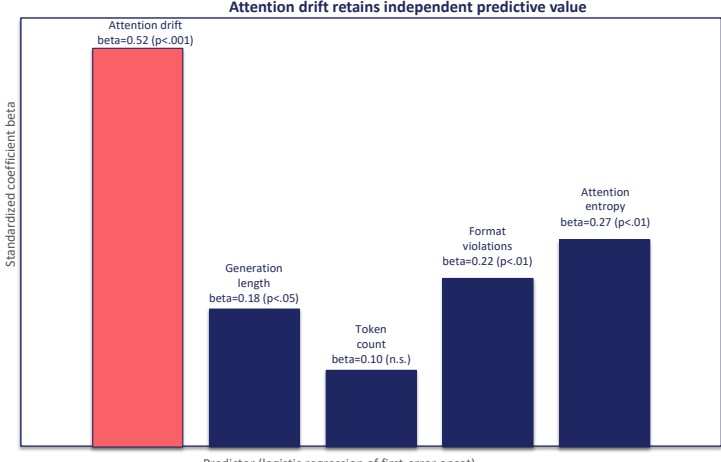

Figure 20: Logistic regression of per-step first-error onset against attention drift, generation length, token count, format violations, and attention entropy. Attention drift retains the largest standardized coefficient ($\beta = 0.52$, $p < 0.001$) after controlling for the other predictors, and adding it to the confound-only model raises AUC from 0.62 to 0.76, an incremental AUC of 0.14. Generation length and format-violation indicators contribute independently but do not subsume the attention signal.

hidden states dropped from 83% to 56%. As a negative control we ablated the same number of heads chosen uniformly at random from each layer and observed a shift from 74% to 71% in valid-transition rates, which was not statistically significant at $p = 0.43$ by the same paired bootstrap. The asymmetry between targeted and random ablation indicates that the state-attending heads form a meaningful functional grouping. The full comparison is shown in Figure 11.

### D.3  Function-Vector Intervention

Following Todd et al. (2024), we added the path-membership function vector to the residual stream at decision-relevant positions, namely the token positions corresponding to the model's enumeration of frontier nodes. We swept the scale parameter $\alpha$ across $\{0.5, 1.0, 1.5, 2.0, 2.5\}$ and report the result at $\alpha = 2.0$, which maximized the behavioral effect without inducing repetition pathologies. Attention to path-member tokens shifted by $8 \pm 3$ percentage points and valid-transition rates rose by $12 \pm 4$ percentage points relative to matched untreated trials. We interpret the modest size of this effect as itself diagnostic: amplification of an already-present direction shifts behavior but cannot substitute for the absent control machinery, consistent with the broader finding that the bottleneck is binding rather than presence. The sweep and the diagnostic reading are shown in Figure 12.

### D.4  Confound Discrimination

To verify that attention drift carries information beyond simpler confounds, we fit a logistic regression predicting per-step first-error onset from attention drift, generation length, token count, format-violation indicators, and attention entropy. Attention drift remained significant ($\beta = 0.52$, $p < 0.001$) after this control, with an incremental AUC of 0.14 above the confound-only baseline of 0.62. Generation length and format-violation indicators contributed independently but did not subsume the attention signal, indicating that attention drift carries information about impending failure that is not reducible to gross properties of the generation process. The standardized coefficient breakdown and the incremental AUC are shown in Figure 20.

# E   Reviewer-Response Checklist

To assist readers in locating revisions made in response to peer review, this appendix maps each concern to the corresponding manuscript section, table, figure, or appendix. The list is grouped by reviewer identifier and is intended as a navigational aid rather than as a self-assessment.

**Causal language and interventions (NBzp).**   Causal phrasing has been tightened throughout the manuscript and is now reserved for sentences that reference the intervention results of Section 5.4; observational sentences use temporal-precedence language instead. The three intervention experiments appear in Section 5.4 with detailed protocols in Appendix D: attention patching (Section 5.4 and Figure 10; Appendix D.1), head ablation (Section 5.4 and Figure 11; Appendix D.2), and the path-membership function-vector intervention (Section 5.4 and Figure 12; Appendix D.3). The confound-discrimination regression appears in Section 5.3, Figure 20, and Appendix D.4. The seven prompting baselines appear in Section 5.1, Table 1, and Figure 3. The model-scale caveat appears in Sections 6 and 6.8, and the deployment subsection in Section 6 has been reframed as directional rather than as a direct safety-critical claim.

**Sensitivity analyses and confound controls (RZsa).**   The state-drift threshold sweep across 0.4 to 0.8 appears in Section 5.3 and Figure 8, alongside alternative reference states (start, goal, random null, middle layer) and a Euclidean-distance variant. The drift reference is defined operationally in Section 4.5, including the layer, token, and aggregation choices, and the metric is explicitly decoupled from ground-truth labels in the same section. RSA confound controls, including node-label randomization, adjacency-list-ordering randomization, the text-distance baseline, and exclusion of hallucinated-node steps, appear in Section 5.2 and Figure 6.

**Protocol auditability, baselines, and presentation (TqXj).**   The Protocol Box, including prompts, parsing rules, the step definition, the first-error definition, and the failure taxonomy, appears in Section 4.2 and Figure 1. The three-step valid-prefix metric, which addresses the short-horizon contrast to the 0% full-path metric, appears in Section 5.1 and Table 1. The RSA sign convention is stated explicitly in Section 4 (RSA subsection) and reiterated in Section 5.2. Notation has been made consistent: $n$ refers to the number of nodes per graph (Section 4.9), trial counts are reconciled with the design as $40 \times 2 \times 3 = 240$ in Section 4.9, and "three independent samples" has been replaced by "three prompt-phrasing variants under deterministic greedy decoding" in Section 4.2. The hybrid-evaluation augmentation with invalid candidates and controlled-reward distractors appears in Section 4 (hybrid subsection), Section 5.1, and Figure 4. The attribution rule for repeated node mentions is stated in Section 4 (attention subsection). The Artifacts and Reproducibility statement appears in Section 4.12.

