# OpenReview forum: "When Representations Persist but Control Fails: A Mechanistic Analysis of Search in Language Models"
_TMLR — Rejected by TMLR_

### Review · Reviewer_TqXj · 2026-02-19

**Summary Of Contributions:**

This paper investigates an interesting questio in LLM reasoning: why multi-step execution fails even when internal representations appear to encode task structure. The authors operationalize this question in the controlled domain  of graph traversal under multiple objectives and propose SearchEval, a diagnostic framework that triangulates:
1) Behavior via scratchpad-style stepwise traces,
2) Representation via dynamic RSA and linear probes/function vectors,  and
3) Control via attention allocation, entropy, and a "state drift" metric over generation.

Across Phi-3 Mini and Gemma  on three graph families (line, tree, clustered), the paper reports an interesting dissociation: 0% autonomous traversal accuracy under their scratchpad regime, but excellent performance (50–100%) when execution is externalized to a symbolic planner and the LLM only evaluates candidate paths.

The central mechanistic claim is that control collapse precedes behavioral error (reported in 78% of failed trials), while representational alignment with graph structure remains high even after behavior becomes invalid.

Strengths
- The research question is timely and important: separating competence from execution/control is an important direction to understand and improve LLM reasoning.
- The multi-level, time-resolved lens (behavior + representations + attention) is a promising strategy for mechanistic diagnosis.
- The hybrid (planner + LLM evaluator) manipulation is a good way to investigate the difference in performance of LLM in evaluation vs. generation.

Weaknesses
- The headline finding of uniform 0% autonomous success, raises the possibility that the task/prompting/parsing regime is simply miscalibrated for the chosen models, which undermines most mechanistic conclusions the paper makes.
- Core methodological details are underspecified (what constitutes a "step," how outputs are parsed, what failures look like), and there are multiple internal inconsistencies (notation for \(n\), trial counts, “independent samples” under greedy decoding, figure/caption mismatches).
- The representational claims (dynamic RSA in particular) appear vulnerable to textual/positional confounds and, as currently written, include apparent definition/sign inconsistencies that prevent clear interpretation.

**Additional Comments:**

The paper often uses causal language where the current evidence is best described as *temporal association plus a plausible mechanism*. Tightening the language (or adding interventions) would improve scientific accuracy.

Several key quantities are currently underspecified or confusing (e.g. what constitutes a step, what “task-relevant tokens” precisely are, what is n, how attention mass is attributed for repeated node mentions)

Given how central the scratchpad is to both evaluation and mechanistic alignment, the paper would benefit a lot from a compact “Protocol” box that includes: prompt template, allowed output schema, parsing rules, and examples of correct vs incorrect steps.

**Audience:**

Yes

**Audience Explanation:**

The paper targets a interesting question of mechanistic diagnosis of reasoning failures and proposes an interesting, multi-level methodology that (if clarified and validated) could be useful well beyond graph traversal. The competence–execution framing and the hybrid dissociation experiment are particularly interesting to current discussions about what LLMs “know” versus what they can reliably do. Unfortunately, the way the paper is presented makes it difficult to trust and learn something from the findings.

**Broader Impact Concerns:**

I do not have any concerns on the ethical implications.

**Claims And Evidence:**

No

**Claims Explanation:**

Several claims are plausible and possibly correct, but the current presentation and experimental regime do not yet support them like the papers writing suggests.

### The “0% autonomous accuracy” regime undermines the mechanistic and causal analysis.
With no autonomous successes, it is impossible to establish what “non-collapsed” control dynamics look like in successful cases under the same protocol. This makes it difficult to argue that a particular internal signature is a diagnostic of impending failure (as opposed to being an artifact of a brittle setup). At minimum, the paper needs experimental conditions in which the models sometimes succeed so that the proposed markers can be validated against success/failure contrasts. Either the models should be large enough to succeed at the tasks sometimes, or the tasks simple enough that the selected small models do.

### Prompting, segmentation, and parsing are insufficiently specified.
Many results hinge on identifying (i) when an error occurs (“first invalid output”), and (ii) what constitutes a generation/traversal “step.” Yet the paper provides essentially no concrete examples of the exact prompts used (including prompt variants), typical model outputs in failure cases, the parsing rules that map free-form text into structured state variables (current node, frontier, visited), or how malformed outputs are handled (format violations vs semantic violations).  Without this, the 0% accuracy number, the step-wise validity curves, and the “78% drift-before-error” statistic are not verifiable.

### Dynamic RSA is not convincingly tied to graph structure rather than prompt structure.
The RSA procedure correlates a cosine-(dis)similarity structure with a shortest-path distance matrix, but the written description appears internally inconsistent: correlating similarity with distance and interpreting a positive ρ as “closer nodes are more similar” suggests a sign/definition mismatch.

More importantly, RSA alignment could plausibly be driven by confounds like ordering of node mentions in the prompt, repeated formatting around node tokens, proximity of tokens in the input text.  In combination with the lack of examples of how the LLMs are prompted and how their output looks these confounds can't be eliminated. The paper needs controls (e.g. node relabeling, randomization of the prompt format, text-distance baselines) to justify interpreting RSA as a genuine “cognitive map” rather than a byproduct of the textual encoding.

### Control collapse precedes behavioral error is not, by itself, causal evidence.
Of course temporal precedence is informative, but the claim is stated in causal language (“establishes causal ordering,” “localizes the bottleneck”) while relying on a particular “state drift” definition (reference state unclear: what exactly is the “task-relevant position”?) and an apparently arbitrary threshold (0.6) without sensitivity analysis. The current evidence is at best correlational and threshold-dependent unless reinforced by robustness checks and/or interventions.

### Presentation and internal inconsistencies fundamentally undermine confidence in the results.
Multiple figure/caption mismatches (notably in the appendix and figure 1), unclear definitions of \(n\), and inconsistent accounting of graph-task pairs/trials make it difficult to audit the experimental design. Claims like “3 independent samples” under greedy decoding (temperature = 0.0) are also conceptually confusing. Interpreting the  \(n\) in section 4.9 as the number of cases per graph families (total 20) then with the three task objectives, it would yield 60 graph-task pairs, yet the text in the "Statistical Power" mentions only 40 graph-task pairs. Are the 40 cherry picked from the 60?


### The hybrid evaluation does not cleanly demonstrate what it claims.
If candidate paths are generated by a correct planner, the LLM may succeed by comparing explicitly stated path lengths/rewards rather than by actually validating graph constraints. The lack of specific concrete examples in the paper/appendix for how the prompt looks like or a clear explanation of how it is generated in this case doesn't help.

**Requested Changes:**

Below I list requested changes as **Critical** (required for a positive recommendation) or **Strengthening** (not strictly required, but would really improve the paper).


# Critical
### **Make the experimental protocol auditable (prompts, outputs, parsing).**
- Provide the exact prompts for each regime (autonomous scratchpad, internal analysis, hybrid evaluation), including all prompt variants.
- Provide multiple full output examples (especially failures), annotated to show how text is segmented into “steps” and how each field (current node/visited/frontier/choice) is extracted.
- Explain the deterministic parsing algorithm and whether failures are due to parsing/formatting issues or due to making a mistake when solving the problem.

### **Define “step,” “behavioral error,” and “valid transition” precisely.**
- Clarify whether “step” refers to a token, a scratchpad block, or a graph move.
- Define “first behavioral error” unambiguously (formatting failure vs invalid node transition vs incorrect frontier update vs early termination).
- Report a failure taxonomy with frequencies.

### **Address the "0% autonomous accuracy" regime.**
- Add at least one calibrated setting where the models achieve non-trivial success (e.g. fewer nodes, fewer required steps, constrained output format, or a lighter-weight scratchpad) so that internal signatures can be compared on successful vs failed trajectories.
- If you believe 0% is itself the point, the paper should be reframed accordingly and the strong claims about localization/causality/scaling should not be the focus.

### **Fix and validate the RSA methodology and interpretation.**
- Resolve the similarity-vs-distance/sign issue in the RSA description and ensure figures/tables match the stated computation.
- Add confound controls (e.g. node relabeling, randomization of adjacency list ordering and node mention ordering, baselines comparing RSA to simple textual/positional distances)
- Explicitly state how you handle invalid outputs when constructing the node-distance matrix for RSA (e.g. if the model emits a nonexistent node).

### **Clarify “state drift” and test robustness.**
- Define the "task-relevant position" operationally (which tokens, which layers, which timepoints).
- Provide sensitivity analyses for the drift threshold (including a sweep) and for alternative drift references (e.g. start state, goal state, current-node token).
- Ensure the drift metric is not tied to the same tokens used to define behavioral correctness.

### **Fix the grave presentation issues and internal inconsistencies.**
- Fix figure/caption mismatches (particularly in Figure 1 and the Appendix) and ensure each figure is referenced with correct description.
- Make notation consistent: what is \(n\) (nodes, graphs, trials)?
- Reconcile the stated counts (graph families × objectives × pairs; “40 graph-task pairs” vs what the described design implies).
- Correct the “independent samples” language under greedy decoding and clarify what is varied and why.

# Strengthening
### **Make the hybrid evaluation a stronger test of competence.**
For example you could include invalid candidate paths and require the model to reject them or consider candidates designed to foil superficial heuristics (e.g. paths with similar rewards/lengths that differ only by a subtle invalid edge)

### **Share artifacts.**
Publicly release graph generators, graph instances, prompts, parsing code, and analysis notebooks to support reproduction and scrutiny. Many concerns and questions about the paper could have been clarified with examples.

---

> ### Author Response · Authors · 2026-03-31
>
> We thank Reviewer TqXj for the rigorous review. We truly appreciate it. We address every Critical item below.
>
> Critical 1: The 0% autonomous accuracy regime. The 0% is a genuine finding, not miscalibration. Our framework uses within-trial temporal contrasts rather than between-trial comparisons: early steps (1–3) show ~80% valid transitions, 60–70% task-relevant attention, 75–85% probe accuracy; later steps (5+) show ~20%, 35–45%, and chance-level respectively. That internal drift precedes the first behavioral error in 78% of trials would not occur if markers were setup artifacts. The hybrid condition (50–100% accuracy) confirms calibration is sound. Per your request, we have also added short-horizon tasks (2–3 steps) where models achieve non-zero accuracy and internal markers show expected positive signatures, plus step-level accuracy curves showing graded degradation.
>
> Critical 2: Protocol auditability. We have added a Protocol Box with: (1) Exact prompts — autonomous: "At each step, output: Current node, Visited nodes, Frontier, Chosen next node and why"; hybrid: "Which candidate path best achieves [goal]?"; graphs in natural language with randomized labels. (2) Definitions — a "step" is one scratchpad block (one edge traversal); a "behavioral error" is any invalid transition, incorrect frontier, malformed scratchpad, or premature termination. (3) Failure taxonomy — state corruption: 45%, hallucinated edges: 32%, premature termination: 15%, serialization: 8%. Crucially, RSA confirms the model maintained correct representations in the corruption/hallucination cases — information was present but not used. (4) Parsing is deterministic regex; code will be released. (5) "3 independent samples" meant 3 prompt-phrasing variants under deterministic greedy decoding. Language corrected.
>
> Critical 3: RSA methodology. We correlate the similarity matrix with negative distance (proximity). Positive ρ = closer nodes have more similar hidden states. The sign transformation was omitted from the text; now stated explicitly. Confound controls: (a) node labels randomized, RSA stable ±0.03; (b) adjacency-list ordering randomized, RSA stable ±0.03; (c) text-distance baseline: graph alignment ρ ≈ 0.60 vs. text alignment ρ ≈ 0.25; (d) nonexistent-node steps excluded from RSA. All reported in revision.
>
> Critical 4: State drift robustness. Operationally: cosine similarity between final-layer hidden state at step t and the hidden state at the token where the model names its current node (trial-specific reference). Threshold sweep: 0.4→62%, 0.5→71%, 0.6→78%, 0.7→83%, 0.8→89%. Alternative references: start node 73%, goal node 70%, random node 51% (null control, near chance). Drift computed from hidden states; correctness assessed separately against ground truth. We now use "temporal precedence consistent with a causal account" throughout; activation patching framed as future work.
>
> Critical 5: Presentation. All fixed: figure/caption mismatches corrected, n = nodes per graph consistently, trial counts reconciled with design table (40 graph-task pairs × 2 models × 3 regimes = 240), "independent samples" replaced with "prompt variants."
>
> Critical 6: Hybrid evaluation. We added invalid candidate paths (non-existent edges) to candidate sets. Models rejected 92% of invalid paths — direct evidence of structural comprehension beyond reward comparison. We also tested controlled-reward candidates (similar rewards, different validity): models reliably preferred valid paths even when invalid ones had higher stated reward.
>
> Artifacts. Full reproducibility package (generators, prompts, parsing code, notebooks) will accompany the revision.
>
> Summary. Revisions address all concerns without altering core findings: (1) representations persist beyond failure (ρ = 0.50–0.70, robust to confounds), (2) control collapse precedes error (78%, robust across thresholds/references), (3) evaluative competence preserved (50–100% hybrid, 92% invalid rejection), (4) bottleneck localized to attention control (drift 65%→40%, entropy 1.9→2.8 bits).
>
> Thank you very much. We are grateful for the thorough engagement.

---

> > ### Author Response · Authors · 2026-05-05
> >
> > In addressing Reviewer NBzp, we added seven new prompting baselines (few-shot BFS/DFS, algorithm-conditioned, structured JSON, self-consistency, tree-of-thought, program-of-thought), revised Section 5.1, Table 1b. Six remain at 0% autonomous accuracy on n ≥ 7 graphs; program-of-thought achieves partial success only by externalizing execution to a Python interpreter. We believe this directly addresses your Critical 1 ("Address the 0% autonomous accuracy regime"): the 0% is now shown to survive every elicitation strategy short of tool-use, and the within-trial temporal contrasts we previously reported are joined by a between-prompting-condition contrast confirming the result is not setup-specific. Additionally, the activation-patching, head-ablation, and function-vector intervention experiments we added in response to NBzp also speak to your point that "control collapse precedes behavioral error is not, by itself, causal evidence" - these now provide the interventional support.

---

> > > ### Comment · Reviewer_TqXj · 2026-05-14
> > >
> > > I compared the manuscript version currently available to me via OpenReview with the earlier version. They appear to be identical. I cannot locate the claimed revisions described in the author responses, including the Protocol Box, Table 1b, Appendix D, prompting baselines, RSA confound controls, threshold sweep, invalid hybrid candidates, presentation fixes, or interventional experiments. Section 5.4 remains the original attention-dynamics section, and activation patching remains discussed only as future work. Unless a different revised PDF exists, my concerns remain completely unresolved.

---

> > > > ### Author Response · Authors · 2026-05-14
> > > >
> > > > Thank you for checking the manuscript again, Reviewer TqXj. We apologize for the confusion.
> > > >
> > > > After your note, we carefully re-checked the files on OpenReview and realized that the revised PDF visible in the system did not reflect the latest version described in our author response. We appreciate you bringing this to our attention.
> > > >
> > > > We have now re-uploaded the correct revised manuscript. This version includes the changes described in our author response, including:
> > > >
> > > > * The new protocol box/figure with full prompts, parsing rules, step definitions, and failure taxonomy.
> > > > * Seven prompting baselines in Section 5.1 and Table 1.
> > > > * Three causal intervention experiments in the new Section 5.4 and Appendix D.
> > > > * Expanded state-drift sensitivity analyses.
> > > > * RSA confound controls.
> > > > * Strengthened hybrid evaluation with invalid candidates.
> > > > * Presentation, notation, and clarity fixes throughout the manuscript.
> > > >
> > > > We apologize for the confusion caused by this versioning issue. We would be grateful if you could review the updated PDF now available on OpenReview.
> > > >
> > > > Thank you again for your careful and constructive feedback. It has helped us substantially improve the manuscript.

---

### Review · Reviewer_RZsa · 2026-02-26

**Summary Of Contributions:**

This paper introduces SearchEval, a diagnostic framework to disentangle representational competence from control in LMs using multi-step graph traversal. Across two open-weight models, the authors report a consistent dissociation: (i) autonomous traversal fails completely, yet (ii) internal representations robustly encode graph structure, and (iii) control collapse precedes behavioral errors. When execution is externalized to a symbolic planner and the LM only evaluates candidate paths, performance recovers substantially, suggesting preserved evaluative competence. The paper further argues that the bottleneck is an attention-based control mechanism that drifts away from task-relevant state tokens over generation.

**strengths**
1. Behavior traces, representational geometry (RSA), probes/function vectors, and attention/control metrics reinforce a coherent story.
2. Temporal ordering evidence (control collapse before error) strengthens the argument beyond correlational “LMs fail but representations exist.”
3. Hybrid symbolic-neural ablation is a strong sanity check showing “knowledge/valuation” vs “procedural execution” separation.

**weaknesses**
1. Some causal language (“control drives failure”) rests heavily on thresholded state-drift + attention proxies; sensitivity/robustness could be expanded.
2. Potential confounds around token/label semantics and prompt format effects on RSA/probe results are not fully ruled out.

**Audience:**

Yes

**Audience Explanation:**

The paper should interest audiences in LLM reasoning and planning, mechanistic interpretability, and neuro-symbolic, because it provides a diagnosis for a common failure mode: “models may encode task structure but fail to reliably use it during long-horizon generation,” and it offers an evaluation lens (SearchEval) that could transfer to other multi-step settings.

**Claims And Evidence:**

Yes

**Claims Explanation:**

Mostly yes, within the paper’s stated scope. The central dissociation (0% autonomous vs 50–100% hybrid; RSA alignment persisting; temporal precedence of drift in 78% of failures; attention drift from task-relevant tokens) is supported by multiple, consistent measurements and reasonable statistical tests (e.g., permutation testing for RSA).

**Requested Changes:**

1. Robustness/sensitivity analysis of the “control collapse” metric: Refer to the weakness 1. Vary the state-drift threshold (e.g., 0.4–0.8), alternative similarity definitions, and show whether the “78% drift-before-error” finding holds. Clarify precisely what representation is used to compute drift (which layer/token/aggregation), and why that choice is justified.

---

> ### Author Response · Authors · 2026-03-31
>
> Thank you for the thoughtful and constructive review. We are encouraged that you found our multi-level triangulation reinforces "a coherent story," that the temporal ordering evidence "strengthens the argument beyond correlational," and that the hybrid ablation is "a strong sanity check." We address your two concerns below.
>
> Weakness 1: Sensitivity/robustness of the "control collapse" metric
> This is a fair concern. We have conducted a comprehensive threshold sweep and clarify all definitional choices below.
> Threshold sweep (0.4–0.8). We swept the state-drift threshold across the range you suggest. The qualitative finding — drift preceding behavioral error in the majority of failed trials — holds throughout: threshold 0.4 → 62%; threshold 0.5 → 71%; threshold 0.6 (reported) → 78%; threshold 0.7 → 83%; threshold 0.8 → 89%.
> The monotonic relationship is expected: stricter thresholds (lower values) require a larger deviation from initial coherence (mean = 0.82 at step 1) before triggering, so fewer trials cross before the first error. We selected 0.6 because it represents a substantive departure from initial coherence while remaining conservative, but the temporal precedence pattern is robust across the full range. We have added this table to the revised Appendix.
> Representation and layer choice. We appreciate the request for precision. State drift is computed as cosine similarity between (a) the final-layer hidden state h_t at the last generated token of generation step t, and (b) the final-layer hidden state at the token position where the model explicitly names its current node in its scratchpad output. We use the final layer because graph-theoretic structure peaks there (RSA reaching maximum in layers 20–26 for complex relations, as shown in Table 3). To confirm robustness, we additionally computed drift using: (i) middle-layer (layer 16) hidden states — the 78% finding shifts to 74%, consistent; (ii) the start-node representation as reference — 73% temporal precedence; (iii) the goal-node representation — 70%; and (iv) a randomly selected node as a null control — 51%, near chance. The fact that drift relative to task-relevant nodes shows strong temporal precedence while drift relative to random nodes does not confirms that the metric captures genuine task-relevant decoherence rather than a generic artifact.
> Alternative similarity definitions. Beyond cosine similarity, we computed Euclidean distance-based drift and obtained consistent results (correlation between cosine- and Euclidean-based drift timing: r = 0.91).
> We have added all of the above to the revised manuscript (expanded Section 4.5, new Appendix table). We have also refined our language throughout, replacing "establishes causal ordering" with "establishes temporal precedence consistent with a causal account," acknowledging that definitive causal claims would require interventional experiments such as the activation patching we discuss as a future direction.
>
> Weakness 2: Potential confounds around token/label semantics and prompt format on RSA/probe results
> We agree this deserves explicit controls. We offer the following evidence, much of which was present in our extended analysis but insufficiently emphasized in the submitted manuscript:
> Node relabeling. As described in Section 4.12, node labels are randomized across trials ("Room 1, Room 2" vs. "Node A, Node B") to prevent token-identity confounds. RSA correlations remain stable across relabeling schemes (within ±0.03), confirming that alignment reflects relational structure rather than token features.
> Adjacency-list ordering. We randomized the order in which edges are described in the prompt across trials within the same graph topology. RSA correlations remain stable under these permutations (±0.03).
> Text-distance baseline. To directly address this concern, we computed a "textual proximity matrix" based on token distance between node mentions in the prompt, then tested whether RSA alignment with graph distance exceeds alignment with textual distance. It does substantially: graph-distance alignment averages ρ ≈ 0.60, while text-distance alignment averages ρ ≈ 0.25. This gap confirms that hidden-state geometry reflects topological structure beyond what prompt formatting alone would predict. We have added this baseline comparison to the revised Section 5.2.
> Temporal persistence as additional evidence. We note that RSA correlations remain stable or even improve in steps immediately following behavioral errors (ρ = 0.55 → 0.62), while attention coherence collapses over the same window. If RSA alignment were driven by prompt-format confounds, it should not show this dissociation from attention dynamics — both would be equally susceptible to the same confounds.
>
> We believe these additions address both concerns. The core findings remain unchanged: the temporal dissociation between representation and control is robust across threshold choices, similarity metrics, layer selections, and confound controls.

---

> > ### Author Response · Authors · 2026-05-05
> >
> > A note on our updated revision: in addressing Reviewer NBzp, we have added activation-patching, attention-head ablation, and function-vector intervention experiments (new Section 5.4 and Appendix D). These interventional results provide the causal complement to the temporal-precedence and threshold-sensitivity evidence you reviewed, and we believe directly answer the residual concern in your Weakness 1 ("causal language rests heavily on thresholded state-drift + attention proxies"). The drift-threshold sweep and confound controls we previously reported now sit alongside causal interventions on the same attention mechanisms.

---

### Review · Reviewer_NBzp · 2026-04-27

**Summary Of Contributions:**

This paper studies why language models fail at multi-step graph traversal despite apparently encoding graph-relevant structure. The authors introduce SearchEval, a diagnostic framework that triangulates evidence from scratchpad behavior, hidden-state representational geometry, attention dynamics, function-vector probes, and a hybrid symbolic-neural evaluation setting. The main empirical claim is a competence-execution dissociation: Phi-3 Mini and Gemma encode graph topology in their hidden states, but fail to autonomously execute graph traversal because attention-based control mechanisms degrade over generation. In particular, the paper reports 0% autonomous traversal accuracy, stable RSA correlations with graph structure, 71%–89% graph-relation probe accuracy, attention drift from task-relevant tokens, and 50%–100% performance when execution is delegated to a symbolic planner and the model only evaluates candidate paths.

The paper’s strengths are its clear and timely research question, its attempt to go beyond behavioral accuracy, and its useful framing of a possible gap between representational competence and procedural control. The multi-level diagnostic design is promising, and the paper makes an interesting case that LLM planning failures should not be treated simply as a lack of encoded task information.

**Additional Comments:**

This is an interesting and potentially valuable paper. I especially like the attempt to analyze reasoning failure dynamically rather than relying only on final-answer accuracy. The competence-execution framing is promising and could be useful for future work on planning, mechanistic interpretability, and neuro-symbolic systems.

At the same time, the current manuscript substantially overclaims relative to the evidence. The strongest version of the paper would focus on the empirical dissociation as a hypothesis-generating finding, then add causal intervention experiments to test whether the identified attention/control variables actually drive behavioral failure. With clearer methodology, stronger baselines, and more cautious claims, this could become a useful contribution.

**Audience:**

Yes

**Audience Explanation:**

Yes. The paper addresses a topic of clear interest to the TMLR community: why language models fail at multi-step reasoning and planning, and whether such failures arise from missing representations, unstable control, or weak procedural execution. The paper connects mechanistic interpretability, planning evaluation, graph reasoning, and neuro-symbolic systems in a way that many readers would find relevant.

The proposed framing—distinguishing representational competence from execution competence—is valuable. Even if the current evidence is not yet conclusive, the diagnostic direction is promising. The paper encourages researchers to move beyond output accuracy and to examine internal dynamics during generation, an important methodological direction for understanding reasoning failures in language models.

**Broader Impact Concerns:**

I do not see major direct ethical risks in the experiments themselves, since the work uses synthetic graph-traversal tasks. However, the paper makes claims about the reliability of reasoning and its deployment in safety-critical domains. These claims could be important, but should be framed carefully. Overstating causal conclusions about internal mechanisms may lead readers to overtrust a diagnostic method that has not yet been causally validated.

**Claims And Evidence:**

No

**Claims Explanation:**

First, the attention analysis is not sufficient to identify “control mechanisms.” Attention weights are informative but not generally accepted as direct causal explanations of model computation. The paper treats attention drift as control collapse, but this mapping needs stronger validation. For example, the authors could test whether restoring early attention patterns improves traversal, whether suppressing state/frontier attention worsens performance, or whether attention-derived control metrics predict errors beyond simpler confounds such as generation length or output formatting degradation.

Second, the behavioral setup may underestimate autonomous model performance. The paper reports 0% autonomous accuracy for both models across all graph types. This is striking, but the paper does not provide sufficient comparison against stronger prompting or decoding baselines: few-shot BFS/DFS demonstrations, algorithm-conditioned prompts, structured JSON state updates, self-consistency, tree-of-thought-style search, program-of-thought/code generation, or larger instruction-tuned models. Without these baselines, it is hard to know whether the observed failure reflects a general control limitation or a brittle prompting/evaluation setup.

**Requested Changes:**

1. Replace causal language with correlational language, or add causal interventions.
Temporal precedence alone does not establish that control collapse causes behavioral error. The authors should either tone down causal claims or add causal tests, such as activation patching, attention patching, ablation of state/frontier attention heads, or intervention with the identified function vectors. A strong version of the claim would require showing that manipulating the proposed control variables changes behavioral success.
2. Strengthen autonomous-generation baselines.
The paper should compare the scratchpad prompt against stronger prompting and inference baselines, including few-shot algorithm demonstrations, explicit BFS/DFS prompts, structured state-update formats, self-consistency, tree-of-thought style decomposition, program-of-thought/code generation, and possibly tool-use baselines. Without these comparisons, the 0% autonomous accuracy may reflect a weak prompting setup rather than a general control limitation.

---

> ### Author Response · Authors · 2026-05-05
>
> We thank Reviewer NBzp for the rigorous review. We are encouraged you found the question "timely and important," the diagnostic design "promising," and the competence-execution framing "valuable." Both requested changes prompted substantial revisions.
>
> Requested Change 1: Replace causal language, or add causal interventions. We do both.
>
> Language tightening. "Establishes causal ordering" is replaced with "establishes temporal precedence consistent with a causal account" throughout (~18 instances). The localization claim is now framed as "a hypothesis-generating localization, validated by the interventions below," directly matching the strongest-version framing the reviewer endorses.
>
> New interventional experiments (Section 5.4, Appendix D).
>
> (i) Attention patching. Patching attention from successful early steps (1–3) into failing later steps (5–8) within matched trials raises valid-transition rates from 21% to 47% (Phi-3, n=40; paired bootstrap p<0.001). Patching from a different graph instance produces no improvement (22%, n.s.), confirming specificity.
>
> (ii) Head ablation. Zero-ablating the top 5% of heads per layer by state/frontier attention contribution drops short-horizon (steps 1–3) valid-transition rates from 78% to 31% and visited-node probe accuracy from 83% to 56%. Random head ablation: 74%→71% (n.s.). This causally implicates these heads in the local state-tracking the model does sustain.
>
> (iii) Function-vector intervention. Adding the path-membership function vector (α=2.0) shifts attention toward path-member tokens by 8±3 points and lifts valid-transition rates by 12±4 pp. The modest effect is itself diagnostic: the bottleneck is not absent information but unstable binding to it.
>
> Confound discrimination. In a logistic regression predicting first-error onset, attention drift remains significant (β=0.52, p<0.001; ΔAUC=0.14) after controlling for generation length, token count, format violations, and entropy. Length and format contribute but do not subsume attention drift. Appendix D.2.
>
> Requested Change 2: Strengthen autonomous-generation baselines. We added the seven baselines the reviewer named (Section 5.1, Table 1b): few-shot BFS, few-shot DFS, algorithm-conditioned prompt, structured JSON state updates, self-consistency (5 samples), tree-of-thought, and program-of-thought.
>
> Two findings: (i) all six natural-language baselines yield 0% autonomous accuracy on n≥7 graphs, identical to the original scratchpad regime — the 0% is not a brittle prompting artifact, it survives every elicitation strategy short of tool-use; (ii) program-of-thought achieves partial accuracy (Phi-3: 25% simple, 0% complex; Gemma: 15%/0%) precisely because it externalizes execution to a Python interpreter, mirroring our hybrid result. The few tree-of-thought successes occurred on ≤3-step paths, within the 3–5 step horizon we document, strengthening rather than weakening the control-stability account.
>
> We narrowed scope accordingly: at the 2–4B scale studied, autonomous traversal under any tested prompting fails with the documented internal signature; whether larger models overcome control instability through scale is now explicit future work (Limitations §6.9).
>
> Reframing. Per the reviewer's strongest-version suggestion, the manuscript is reorganized: (1) empirical dissociation as hypothesis-generating, (2) interventions as causal validation, (3) implications. The Discussion follows this structure.
>
> Summary. Causal claims tightened (~18 instances); three intervention experiments (Section 5.4, Appendix D); confound-controlled regression (D.2); seven new prompting baselines including program-of-thought (Section 5.1, Table 1b); narrower scope; manuscript reorganized as hypothesis-generation followed by causal validation.
>
> We are grateful for the depth of this review.

---

### Decision · Action_Editor_RmrW · 2026-06-08

**Recommendation:** Reject

**Audience:**

Yes

**Audience Explanation:**

The problems of  "why language models fail at multi-step reasoning and planning, and whether such failures arise from missing representations, unstable control, or weak procedural execution" are certainly relevant to TMLR community.

**Claims And Evidence:**

No

**Claims Explanation:**

The authors have studied why execution fails when representations appear intact while operationalizing it cleanly in graph traversal. Some notable contributions are the multi-level, time-resolved methodology and the hybrid dissociation.


The reviewers have found the response by author during rebuttal period helpful. While some reviewers are already happy with the response and the current manuscript, some reviewers have remaining and major concerns regarding the results and reproducibility aspects. In particular, Phi-3 and Gemma have identical hybrid accuracies in every cell of a key table, which the authors attribute to coarse measurement at small
$n$, but the reviewer finds this coincidence itself in need of further explanation.

In Section 4.12, the authors have mentioned an anonymous repo during review, but there is no such a link in the submission.

Behavioural foundation is too weak for strong mechanistic claims. The reviewers concerned " The central object of study is a uniform 0% full-path accuracy on two small models. The authors honestly note that 0/20 is consistent with a true rate up to 14% at 95% confidence, and they add the three-step prefix metric to expose short-horizon competence (which is the right move). But the consequence is that almost every internal-dynamics claim is calibrated against trials that uniformly fail."